# Don't Paint Everyone with the Same Brush: Adaptive Prompt Prototype Learning for Vision-Language Models

## Abstract

Vision Language Models (VLMs) have demonstrated great potential on zero-shot classification tasks by computing the similarity between visual and textual embeddings. To adapt VLMs to a downstream task, recent advances introduced context optimization. It learns a single embedding for either visual or textual modalities, aiming to improve performance on both base and new classes. However, we identify a critical issue by using single embedding for each class. That is, for image samples of a single class, the visual appearance may vary significantly. Thus, existing methods relying on a singular textual embedding fail to capture the visual variance, leading to suboptimal performance on downstream tasks. In this paper, we propose an Adaptive Prompt Prototype Learning (APPLe) for VLMs. Specifically, we build various prompts as class prototypes to cover the visual variance. Moreover, there are inevitably some ambiguous words in prompts, bringing noise to the textual features. To resolve this problem, an adaptive attention mechanism is designed to weigh the importance of different prototypes. It learns to assign higher scores to the representative prototypes, and lower scores to the flawed or less representative prototypes. To evaluate the effectiveness of APPLe, we conduct experiments on three representative tasks, *i.e.,* generalization to unseen classes, new target datasets, and unseen domain shifts. APPLe exhibits a consistent performance improvement of 3.66% on new classes and 2.79% on the harmonic mean.

## 1 Introduction

In real-world scenarios, visual recognition grapples with high variability and intricate nuances. We may consider a seemingly simple visual category: apple pies. As illustrated in Fig. 1, while they all belong to the "apple pie" class, their presentations vary drastically in terms of color, texture, shape, background, and even serving style. This rich visual variance challenges the simple prompt template: "a photo of an apple pie". Whereas the template seems broad and generic, apple pies in these images can span from a round, golden-crusted masterpiece to a homemade pie adorned with a delicate lattice crust. Therefore, a singular prompt, like "a photo of an apple pie", may not be able to encapsulate the myriad visual intricacies that these images present.

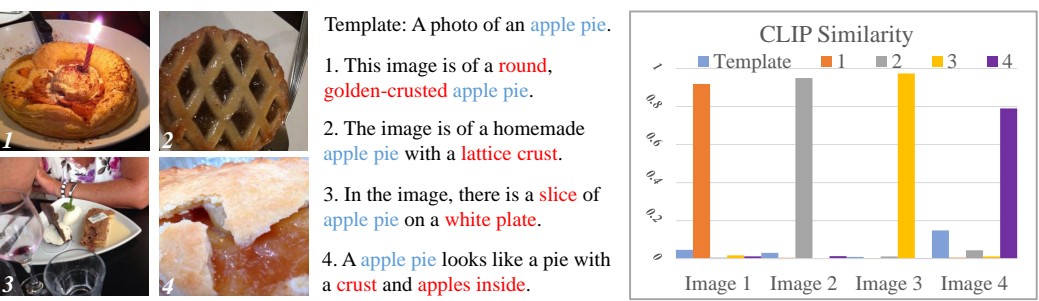

Template: A photo of an apple pie.

1. This image is of a round, golden-crusted apple pie.

2. The image is of a homemade apple pie with a lattice crust.

3. In the image, there is a slice of apple pie on a white plate.

4. A apple pie looks like a pie with a crust and apples inside.

Figure 1: Left: images and prompts of apple pies. Right: similarities between images and prompts.

Large Vision-Language Models (VLMs) like CLIP (Radford et al., 2021) and Align (Jia et al., 2021) have made strides in zero-shot classification by leveraging relevant text prompts for testing an image. These prompts, typically prefixed with "a photo of a," paired with category names, have shown promise in many benchmark datasets. Existing prompt-tuning methods (Zhou et al., 2022b;a; Khattak et al., 2023) push forwards in-context learning, which optimizes the prefix embedding in either text or visual modalities. It can not only improve the performance of the base classes, but also improve the generalization ability to new classes without any visual samples.

However, the crux remains in real-world visual data. Due to the uncontrollable nature of visual data generation, they are drenched in variance. It is not just about recognizing an apple pie; it is about discerning its myriad presentations and subtleties. A simple prompt template can demonstrate strong relevance to the category, but might not encapsulate this vast diversity. This disparity becomes even more evident when we compare images with their varied textual descriptions, as shown in Fig. 1. The level of similarity between an image and its detailed description often surpasses its similarity to a generic category name. There is an intricate connection between class names and descriptive keywords. While an image might resonate strongly with a combination of both, it may not connect as powerfully with the class name only.

Borrowing from the adage "**Don't paint everyone with the same brush**", our approach discards the way of using the same prompt template for every sample within a category. Instead, to encapsulate this large visual variance, we take advantage of multiple prompt prototypes. There are diverse prompts constructed by approaching an object from multiple angles. In this way, even if an image sways from a typical visual example of its category, the rich textual keywords in the prompt can guide through the classification process.

However, we also notice that the prompts inevitably involve ambiguous and flawed keywords that are detrimental to decision-making. To mitigate this issue, we introduce an adaptive attention mechanism. It learns to lower the confidence for such prompts, while assigning higher attention values to those more accurate and representative prototypes in a given context. In addition, it is unlikely for a visual sample to align closely with every prototype. We thus introduce a prototype decorrelation penalty to minimize the probability of the co-occurrence of multiple prototypes.

We show that, without fine-tuning any parameters like context optimization methods, zero-shot CLIP with prompt prototypes can achieve better performance on unseen classes. The result challenges the generalization ability claimed in existing context optimization methods. On the other hand, we confirm that the original CLIP already possesses strong generalization ability on new classes, but a singular prompt cannot exploit such ability. Specifically, we conducted experiments on 11 datasets across different settings. On few-shot learning setting, APPLe can perform better than state-of-the-art method PLOT (Chen et al., 2022) by 1.86%, 1.0%, 0.50%, 0.51%, and 0.12% PLOT at 1, 2, 4, 8, and 16 shots. Our training-free version can achieve an averaged 3.83% performance gain on the new classes on the state-of-the-art method MaPLe (Khattak et al., 2023). The training-free version does not fine-tune any model parameters nor use any image samples as a support set. As for trainable settings, our experimental results demonstrate consistent improvement over MaPLe on 11 datasets when generalizing to new classes, new target datasets, and new domains.

## 2 RELATED WORK

**Vision Language Models**    Inspired by the success of large-scale language models like BERT (Devlin et al., 2018) and GPT series (Radford et al., 2018; 2019; Brown et al., 2020; OpenAI, 2023) in NLP, researchers began pre-training VLMs on large datasets to then fine-tune them on downstream tasks (Su et al., 2019; Tan & Bansal, 2019). Recent trends in the field lean towards a unified model for vision and language that can be jointly trained on multiple tasks. The CLIP (Radford et al., 2021) model learns visual and linguistic representations in a zero-shot manner by training on a large set of images paired with natural language descriptions. Similarly, ALIGN model (Jia et al., 2021) pushes the boundary by scaling up the data and the model size, achieving state-of-the-art performance on multiple benchmarks. Florence (Yuan et al., 2021) further extends the representations to fine-grained objects, videos, and multiple modalities such as caption and depth. Although these pre-trained VLMs have learned generalized representations for both vision and languages, adapting to downstream tasks remains a challenging research problem. There have been many tailored methods proposed to adapt VLMs for few-shot classification (Gao et al., 2021; Kim et al., 2021; Zhang

et al., 2021), object detection (Feng et al., 2022; Gu et al., 2021; Maaz et al., 2022), and semantic segmentation (Ding et al., 2022; Li et al., 2021; Lüddecke & Ecker, 2022).

**Prompt Engineering**  In VLMs, a text encoder can embed the hand-crafted prompts (Wang et al., 2022) constructed by the category names into a common space shared with visual features, enabling zero-shot visual recognition. To adapt to downstream tasks, CoOp (Zhou et al., 2022b) presents a context optimization method that fine-tunes a continuous set of prompt vectors in the text encoder. Co-CoOp (Zhou et al., 2022a) shows CoOp can overfit to the base classes, leading to inferior performance on new classes. They instead learn a context vector conditioned on the visual samples to solve the overfitting issue. ProDA (Lu et al., 2022) is proposed to learn the distribution of prompts by optimizing multiple sets of prompts. PLOT (Chen et al., 2022) apply optimal transport to match the vision and text modalities for prompt learning. Instead of optimizing the context vector for textual prompts, VisPro (Bahng et al., 2022) presents a visual prompting method that learns an image perturbation to adapt to VLMs. Along the same line, MaPLe (Khattak et al., 2023) proposes to optimize the context information in both vision and text modalities to improve the connection between the visual and textual representations. These methods have a common constraint on learning a single context vector to cover the entire downstream tasks. We instead leverage multiple prompt prototypes to reflect the visual variance from various angles. Menon & Vondrick (2022) and Pratt et al. (2023) also explore the use of GPT-generated prompts for visual classification. While there are similarities in leveraging GPT for prompt generation, our approach diverges in several key aspects. Firstly, unlike their idea of leveraging the prompts for zero-shot CLIP inference, we further fine-tune these prompts and provide the in-context ability for downstream tasks. In conclusion, while there are thematic overlaps with (Menon & Vondrick, 2022; Pratt et al., 2023), our approach introduces an adaptive prompt prototype learning method for better leveraging GPT-3 prompts for VLMs, adding valuable insights and alternatives to the current body of research on visual classification using language models.

**Prototype Learning**  Prototype learning traces its roots to classical models such as K-Nearest Neighbors (Peterson, 2009) and Learning Vector Quantization (Kohonen & Kohonen, 1995). These methods inherently rely on instance-based learning, where decisions are based on the proximity to a set of representative examples or prototypes. Deep models, such as Prototypical Networks for few-shot learning (Snell et al., 2017), learn a metric space in which classification can be performed by computing distances to prototype representations of each class. These models have demonstrated strong performances, especially in tasks with limited labeled data. Recognizing that a single prototype might not capture the entire variance within a class, there are methods to construct multiple prototypes for pathology (Deuschel et al., 2021), face recognition (Zhao et al., 2019), semantic segmentation (Yang et al., 2020; Sacha et al., 2023). This enables capturing diverse modes within each category, leading to more nuanced and accurate recognition. Yet, a significant constraint arises when these prototypes are constructed solely from visual data. In a few-shot scenario, the quality and diversity of prototypes become limited to the number of available samples within each class. Consequently, if class samples have a skew towards a particular style or attribute, these multi-prototype models may fail to capture the genuine variance of the class. This limitation serves as our motivation. In this work, we investigate beyond visual samples, harnessing the capabilities of VLMs. By sourcing multiple prototypes directly from textual descriptions, we manage to paint a more comprehensive and varied picture of visual categories, resulting in both diverse and accurate prototypes.

## 3 PRELIMINARIES

Prompt engineering leverages the versatility of VLMs (CLIP in this work) to perform zero-shot classification. In essence, class names are combined with manually tailored textual templates to produce prompts. Consider $C$ class names, represented as $\texttt{class}_c$, $c \in \{1, \ldots, C\}$. Each class name is placed within a template, generating a prompt $\boldsymbol{h}_c = \texttt{a photo of a } \{\texttt{class}_c\}$. The CLIP text encoder processes this prompt $\texttt{TextEnc}(\cdot)$, and compute the class-specific text feature $\boldsymbol{t}_c = \texttt{TextEnc}(\boldsymbol{h}_c)$. For any image $\boldsymbol{x}$ to be classified, they are passed through the image encoder, $\texttt{ImageEnc}(\cdot)$, resulting in image features $\boldsymbol{f} = \texttt{ImageEnc}(\boldsymbol{x})$. The image class probabilities are then computed by the cosine similarity between its visual features and the text features, normalized

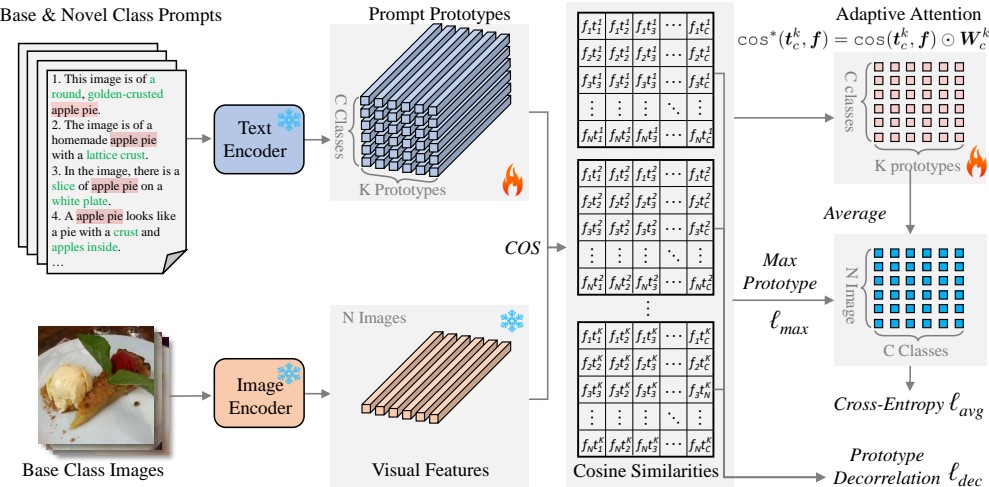

Figure 2: An illustration of APPLe for VLMs. We calculate the cosine similarities between the prompt prototypes and the visual features. The adaptive attention matrix is introduced to weight the prototype logits. The maximum prototype logits are used to calculate $\ell_{max}$. For the prototype logits within one class, we apply a decorrelation loss $\ell_{dec}$ to suppress the co-occurrence of prototypes. The averaged class logits are used to calculate $\ell_{avg}$.

by a temperature factor $\tau$:

$$P(y|\boldsymbol{x}) = \frac{\exp\big(\cos(\boldsymbol{t}_y, \boldsymbol{f})/\tau\big)}{\sum_{c=1}^{C} \exp\big(\cos(\boldsymbol{t}_c, \boldsymbol{f})/\tau\big)}. \tag{1}$$

Subsequently, the class label of the image $\boldsymbol{x}$ is deduced as $\tilde{y} = \text{argmax}_y P(y|\boldsymbol{x})$.

## 4 ADAPTIVE PROMPT PROTOTYPE LEARNING

**Prompt Construction**    To acquire precise, articulate, and high-quality prompts, we leverage the pre-trained large language model GPT-3 (Brown et al., 2020). For each class in our downstream tasks, GPT-3 generates an array of richly varied descriptions, ensuring broad coverage of potential visual interpretations for each category. Taking the example illustrated in Figure 1, descriptions of an apple pie capture disparate visual facets. Notably, certain descriptors, such as `round` and `slice`, are mutually exclusive in the context of visualizing apple pies. Suppose we acquire $K$ prompts for $C$ classes, $\{\boldsymbol{h}_c^k\}_{c=1,k=1}^{C,K}$, where each $\boldsymbol{h}_c^k$ corresponds to the $k^{th}$ prompt for the $c^{th}$ class.

**Prompt Prototype Learning**    In traditional supervised learning, the prototype for a given class is typically represented by the mean vector of the visual features. However, with the advent of VLMs, this paradigm has been reshaped. In the context of VLMs, text features derived from prompts can serve as an arbitrary number of prototypes, offering a more semantic perspective on class representation. Given K prompts, the averaged class logits over multiple prototypes can be represented as $\sum_{k=1}^{K} \cos(\boldsymbol{t}_y^k, \boldsymbol{f})/K$. The main advantage of this approach lies in the diverse textual descriptions of the prompts. CLIP adeptly fuses visual and semantic modalities. The diversity ensures a broader coverage of visual variance. This is particularly valuable, especially when visual samples are biased or limited.

**Adaptive Attention**    Apart from the benefits derived from the diversity of prompt prototypes, our approach also acknowledges the presence of ambiguous and flawed descriptors within the prompts. To mitigate the impact of these flawed descriptors, our model is designed to assign lower confidence to such prompts while amplifying the influence of more accurate and representative ones.

To this end, we introduce adaptive attention to consider the varying significance among the diverse prototypes. Given prototypes across $C$ classes, each with $K$ prompts, we formulate an attention matrix $\boldsymbol{W} \in \mathbb{R}^{C \times K}$. This matrix is employed to weigh the cosine similarities as follows:

$$\cos^*(\boldsymbol{t}_c^k, \boldsymbol{f}) = \cos(\boldsymbol{t}_c^k, \boldsymbol{f}) \odot \boldsymbol{W}_c^k, \quad \boldsymbol{W} \in \mathbb{R}^{C \times K}. \tag{2}$$

When averaging the cosine similarities of prototypes within a class, the representative ones might be smoothed out by others. Thus, allocating a higher attention score to the representative prototypes is crucial to preserving their influence on the averaged result.

**The Closest Prototype** We assume that every visual sample will exhibit the highest degree of similarity to one specific prompt prototype. To take advantage of these optimal visual-prompt pairs, we exclusively consider the logits corresponding to the prompt prototypes that are the closest to the visual samples for loss computation. We design a maximum cross-entropy loss as follows:

$$\ell_{max} = -log \frac{\exp(\cos^*(\boldsymbol{t}_y^{\tilde{k}}, \boldsymbol{f})/\tau)}{\sum_{c=1}^{C} \exp(\cos^*(\boldsymbol{t}_c^{\tilde{k}}, \boldsymbol{f})/\tau)}, \quad \tilde{k} = \arg\max_k \cos^*(\boldsymbol{t}_y^k, \boldsymbol{f}), \tag{3}$$

where $\tilde{k}$ is the index of the prototype with maximum cosine similarity. This approach ensures that the primary emphasis is placed on optimizing the relationships between visual samples and the most congruent prompt prototypes.

**Prototype Decorrelation** Our method ensures diversity among the constructed prototypes within each class, acknowledging the inherent intra-class variance. Given this variance, it is unlikely for a visual sample to align closely with every prototype. To mitigate the cumulative impact of many prototypes, we integrate a decorrelation loss:

$$\ell_{dec} = \sum_{c=1}^{C} \sum_{k=1}^{K} \left\| \cos^*(\boldsymbol{t}_y^k, \boldsymbol{f}) \right\|_2, \tag{4}$$

where a $\ell_2$ norm is adopted to reduce the magnitude of the summed logits within each class, aiming to suppress the likelihood of co-occurrence of multiple prototypes and enhance the modal's ability to discriminate different visual instances effectively.

**Training** The overall training process of APPLe is presented in Fig. 2. The prompt prototypes are initialized by extracting the textual features from the prompts with the text encoder. In APPLe, the trainable parameters include the prompts prototypes, $\boldsymbol{T} = \sum_{k=1}^{K} \sum_{c=1}^{C} \boldsymbol{t}_c^k, \boldsymbol{T} \in \mathbb{R}^{C \times K \times D_t}$, along with the attention matrix $\boldsymbol{W} \in \mathbb{C} \times \mathbb{K}$. In addition to the above two loss functions, we consider the prediction logits from all prototypes and take the average logits to calculate the cross-entropy loss:

$$\ell_{avg} = -log \frac{\exp\left( \left( \sum_{k=1}^{K} \cos^*(\boldsymbol{t}_y^k, \boldsymbol{f}) \right)/(\tau * K) \right)}{\sum_{c=1}^{C} \exp\left( \left( \sum_{k=1}^{K} \cos^*(\boldsymbol{t}_c^k, \boldsymbol{f}) \right)/(\tau * K) \right)}. \tag{5}$$

Therefore, the overall training objective is:

$$\ell_{overall} = \ell_{avg} + \lambda_1 \ell_{max} + \lambda_2 \ell_{dec}, \tag{6}$$

where $\lambda_1$ and $\lambda_2$ are consistently set to 3 for all experiments in all datasets. It is worth mentioning that fine-tuning the textual features typically poses the overfitting issue to base classes, which significantly degenerates the generalization ability to new classes. However, APPLe with multiple prototypes can mitigate this issue, and instead enhance the generalization ability.

**Inference** At test time, the class distribution of a test image can be computed as follows:

$$P(y|\boldsymbol{x}) = \frac{\exp\left( \left( \sum_{k=1}^{K} \cos^*(\boldsymbol{t}_y^k, \boldsymbol{f}) + \beta \cos^*(\boldsymbol{t}_y^{\tilde{k}}, \boldsymbol{f}) \right)/(\tau * K) \right)}{\sum_{c=1}^{C} \exp\left( \left( \sum_{k=1}^{K} \cos^*(\boldsymbol{t}_c^k, \boldsymbol{f}) + \beta \cos^*(\boldsymbol{t}_c^{\tilde{k}}, \boldsymbol{f}) \right)/(\tau * K) \right)}, \tag{7}$$

where we consider the weighted average cosine similarities of the prototypes of every class and also pick the maximum cosine similarity from the closest prototype. $\beta$ controls the weight of the closest prototype. The class label of image $\boldsymbol{x}$ is then deduced as $\tilde{y} = \arg\max_y P(y|\boldsymbol{x})$.

Notably, without training the prototypes, the formulation in Eq. 7 offers a distinct zero-shot classification approach compared to Eq. 1. This training-free method simply uses the $k$ prompts to conduct classification on Base and New classes, which neither requires fine-tuning any CLIP components nor using image samples as a support set.

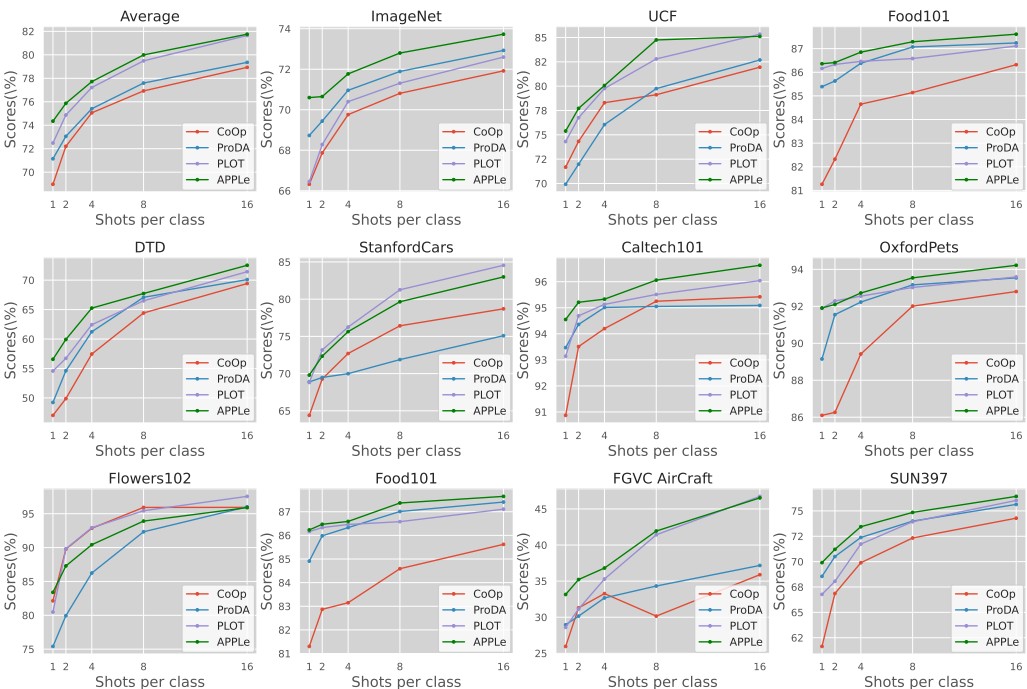

Figure 3: The few-shot learning results on 11 datasets. We compare our APPLe with CoOp, ProDA, PLOT and observe the consistent and significant performance improvement on most datasets. (The average accuracy on all datasets is shown on the left top.)

## 5 EXPERIMENTS

**Baselines and Datasets** We quantitatively and qualitatively compare our methods against CoOp (Zhou et al., 2022b), Co-CoOp (Zhou et al., 2022a), ProDA (Lu et al., 2022), PLOT (Chen et al., 2022) , and MaPLe (Khattak et al., 2023). These are the most recently established methods for adapting CLIP to downstream object recognition tasks. To analyze the capabilities of our approach on real images, we consider a variety of image domains, including a scene recognition dataset SUN397 (Xiao et al., 2010); an action recognition dataset UCF101 (Soomro et al., 2012); a satellite image dataset EuroSAT (Helber et al., 2019); a texture dataset DTD (Cimpoi et al., 2014); two coarse-grained object datasets, ImageNet (Deng et al., 2009) and Caltech101 (Fei-Fei et al., 2004) and five fine-grained datasets, OxfordPets (Parkhi et al., 2012), StanfordCars (Krause et al., 2013), Flowers102 (Nilsback & Zisserman, 2008), Food101 (Bossard et al., 2014), and FGVCAircraft (Maji et al., 2013). More information about the datasets, the generated prompts, and the implementation details are given in the Appendix.

**Few-Shot Learning Results** We compare our methods with CoOp, ProDA and PLOT on the few-shot learning setting. The results are summarized in Figure 3, where the green line denotes our APPLe method, the red line is the CoOp, blue line is ProDA and the purple line represents PLOT. Taking the average accuracy (at the left top) as the example, PLOT respectively gained 1.86%, 1.0%, 0.50%, 0.51%, and 0.12% performance boost over PLOT at 1, 2, 4, 8, and 16 shots. It can be seen that APPLe performs particularly well on very low shots. The numeric results are reported in Section A.7

**Training-Free Results** As shown in Table 1, the method APPLe* represents the training-free version, in which there are no training samples observed by the model or used as support samples. It is worth mentioning that *the average performance in new classes surpasses all existing training-based methods*. Overall, we can achieve an average accuracy on HM of 74.83% on 11 datasets. Particularly, the performance gain over zero-shot CLIP on the texture dataset DTD is most signifi-

Table 1: Comparison with state-of-the-art methods on base-to-new generalization. APPLe demonstrates strong generalization results over existing methods on 11 recognition datasets. * represents the training-free version of APPLe. Absolute performance improvements are indicated in blue.

| Dataset | Set | Training-free Methods | | | Training-based Methods | | | | |
| | | CLIP | APPLe* | Δ | CoOp | Co-CoOp | MaPLe | APPLe | Δ |
|---|---|---|---|---|---|---|---|---|---|
| Average | Base | 69.34 | 72.21 | +2.87 | 82.69 | 80.47 | 82.28 | 84.26 | +1.98 |
| | New | 74.22 | 78.05 | +3.83 | 63.22 | 71.69 | 75.14 | 78.80 | +3.66 |
| | HM | 71.70 | 74.83 | +3.13 | 71.66 | 75.83 | 78.55 | 81.34 | +2.79 |
| ImageNet | Base | 72.43 | 74.62 | +2.19 | 76.47 | 75.98 | 76.66 | 78.17 | +1.51 |
| | New | 68.14 | 71.94 | +3.80 | 67.88 | 70.43 | 70.54 | 72.12 | +1.58 |
| | HM | 70.22 | 73.26 | +3.04 | 71.92 | 73.10 | 73.47 | 75.02 | +1.55 |
| Caltech101 | Base | 96.84 | 96.06 | -0.78 | 98.00 | 97.96 | 97.74 | 98.26 | +0.52 |
| | New | 94.00 | 95.74 | +1.74 | 89.81 | 93.81 | 94.36 | 95.63 | +1.27 |
| | HM | 95.40 | 95.90 | +0.50 | 93.73 | 95.84 | 96.02 | 96.93 | +0.91 |
| OxfordPets | Base | 91.17 | 93.46 | +2.29 | 93.67 | 95.20 | 95.43 | 95.64 | +0.21 |
| | New | 97.26 | 97.99 | +0.73 | 95.29 | 97.69 | 97.76 | 98.04 | +0.28 |
| | HM | 94.12 | 95.67 | +1.55 | 94.47 | 96.43 | 96.58 | 96.83 | +0.25 |
| Stanford Cars | Base | 63.37 | 64.49 | +1.12 | 78.12 | 70.49 | 72.94 | 80.23 | +7.29 |
| | New | 74.89 | 75.79 | +0.90 | 60.40 | 73.59 | 74.00 | 75.12 | +1.12 |
| | HM | 68.65 | 69.68 | +1.03 | 68.13 | 72.01 | 73.47 | 77.59 | +4.12 |
| Flowers102 | Base | 72.08 | 75.02 | +2.94 | 97.60 | 94.87 | 95.92 | 96.58 | +0.66 |
| | New | 77.80 | 80.35 | +2.55 | 59.67 | 71.75 | 72.46 | 78.58 | +6.12 |
| | HM | 74.83 | 77.59 | +2.76 | 74.06 | 81.71 | 82.56 | 86.66 | +4.10 |
| Food101 | Base | 90.10 | 90.37 | +0.27 | 88.33 | 90.70 | 90.71 | 90.99 | +0.28 |
| | New | 91.22 | 91.68 | +0.46 | 82.26 | 91.29 | 92.05 | 91.88 | -0.17 |
| | HM | 90.66 | 91.02 | +0.36 | 85.19 | 90.99 | 91.38 | 91.43 | +0.05 |
| FGVC AirCraft | Base | 27.19 | 30.07 | +2.88 | 40.44 | 33.41 | 37.44 | 44.66 | +7.22 |
| | New | 36.29 | 41.15 | +4.86 | 22.30 | 23.71 | 35.61 | 43.13 | +7.52 |
| | HM | 31.09 | 34.75 | +3.66 | 28.75 | 27.74 | 36.50 | 43.88 | +7.38 |
| SUN397 | Base | 69.36 | 74.57 | +5.21 | 80.60 | 79.74 | 80.82 | 82.44 | +1.62 |
| | New | 75.35 | 78.17 | +2.82 | 65.89 | 76.86 | 78.70 | 79.04 | +0.34 |
| | HM | 72.23 | 76.33 | +4.10 | 72.51 | 78.27 | 79.75 | 80.70 | +0.95 |
| DTD | Base | 53.24 | 63.54 | +10.30 | 79.44 | 77.01 | 80.36 | 82.41 | +2.05 |
| | New | 59.90 | 67.27 | +7.37 | 41.18 | 56.00 | 59.18 | 69.57 | +10.39 |
| | HM | 56.37 | 65.35 | +8.98 | 54.24 | 64.85 | 68.16 | 75.45 | +7.29 |
| EuroSAT | Base | 56.48 | 57.31 | +0.83 | 92.19 | 87.49 | 94.07 | 90.90 | -3.17 |
| | New | 64.05 | 78.18 | +14.13 | 54.74 | 60.04 | 73.23 | 81.69 | +8.46 |
| | HM | 60.03 | 66.14 | +6.11 | 68.69 | 71.21 | 82.35 | 86.05 | +3.7 |
| UCF101 | Base | 70.53 | 74.77 | +4.24 | 84.69 | 82.33 | 83.00 | 86.56 | +3.56 |
| | New | 77.50 | 80.31 | +2.81 | 56.05 | 73.45 | 78.66 | 81.99 | +3.33 |
| | HM | 73.85 | 77.44 | +3.59 | 67.46 | 77.64 | 80.77 | 84.21 | +3.44 |

cant, achieving 8.98%. The overall performance is comparable to the training-based methods CoOp and Co-CoOp, which demonstrates the effectiveness of prototypes.

**Generalization from Base to New Classes** Table 1 presents the performance comparison with baseline methods in a base-to-new generalization setting. The evaluation is conducted on the base and new classes separately to test generalizability. After fine-tuning the prompt prototypes and learning the attention matrix on the base classes, we achieved significant performance gains on both base and new classes. On the 11 datasets, we consistently outperform baseline methods on HM. The average absolute gain on HM is 2.79%. In particular, our performance improvement on ImageNet is 1.55%, which is a significant improvement considering the 1,000 classes in ImageNet. In addition, the performance of FGVC AirCraft and DTD are significantly improved from 31.09% to 43.88% and 68.16% to 75.45%, respectively. The great improvement showcases the ***strong adaptation ability to those under-presentative classes in the original training set***.

**Impact of Prototype Number** In Fig. 4, we study the impact of the number of prototypes on the performance using the ImageNet dataset. The solid line represents the performance trend of APPLe. The dashed line is the zero-shot CLIP performance, which provides a baseline comparison to APPLe. As we vary the number of prototypes from 1 to 50, there is an obvious up-

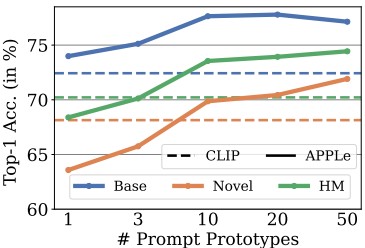

Figure 4: Performance comparison of APPLe with various numbers of prototypes on ImageNet dataset.

trend in performance for both base and new classes. The finding highlights our core motivation that a singular prompt cannot cover the entire variance space of a visual category, but employing a diverse range of prompt prototypes is instrumental in encompassing the extensive variance. Notably, while fine-tuning with one prompt prototype, the performance drops on new classes and HM. This observation confirms that fine-tuning one prompt will lead to overfitting to base classes. However, when we use more than 3 prototypes, the performance of APPLe on new classes surpasses zero-shot CLIP. Thus, *we confirm that the overfitting problem can be mitigated by using more prompt prototypes.*

**Domain Generalization**  The effectiveness of APPLe in generalizing to out-of-distribution datasets is demonstrated when compared to methods such as CoOp, Co-CoOp, and MaPLe. We evaluate the direct generalization ability of a model trained on ImageNet to several out-of-domain datasets, including ImageNetV2, ImageNet-Sketch, ImageNet-Adversarial,

Table 2: Comparison of APPLe and baselines on domain generalization. All methods are trained on 16 images per class from 1,000 classes on ImageNet.

|  | Source | Target | | | |
|---|---|---|---|---|---|
|  | ImageNet | ImageNetV2 | ImageNet-S | ImageNet-A | ImageNet-R |
| CLIP | 66.73 | 60.83 | 46.15 | 47.77 | 73.96 |
| CoOp | 71.51 | 64.20 | 47.99 | 49.71 | 75.21 |
| Co-CoOp | 71.02 | 64.07 | 48.75 | 50.63 | 76.18 |
| MaPLe | 70.72 | 64.07 | 49.15 | 50.90 | 76.98 |
| APPLe* | 69.88 | 63.31 | 49.18 | **50.92** | **77.09** |
| APPLe | **72.32** | **65.07** | 49.38 | 50.80 | **77.09** |

and ImageNet-Rendition. The results, presented in Table 2, showcase that APPLe outperforms existing methods on both source domain and target domains, except on ImageNet-Adversarial. The results confirm that *APPLe is more domain-generalizable than context optimization methods.*

Table 3: Ablation study of APPLe on ImageNet.

| Components | | | | | Performance | | |
|---|---|---|---|---|---|---|---|
| Prototypes | Training | Attention | $\ell_{max}$ | $\ell_{dec}$ | Base | New | HM |
|  |  |  |  |  | 72.43 | 68.14 | 70.22 |
| ✓ |  |  |  |  | 74.68 | 71.88 | 73.25 |
| ✓ |  | ✓ |  |  | 75.46 | 72.07 | 73.73 |
|  | ✓ |  |  |  | 73.84 | 63.57 | 68.31 |
| ✓ | ✓ |  |  |  | 76.12 | 72.04 | 74.02 |
| ✓ | ✓ | ✓ |  |  | 76.25 | 72.09 | 74.11 |
| ✓ | ✓ |  | ✓ |  | 77.40 | 72.07 | 74.64 |
| ✓ | ✓ |  |  | ✓ | 76.14 | 72.09 | 74.06 |
| ✓ | ✓ | ✓ | ✓ |  | 77.82 | 72.04 | 74.82 |
| ✓ | ✓ |  | ✓ | ✓ | 77.88 | 72.09 | 74.87 |
| ✓ | ✓ | ✓ | ✓ | ✓ | 78.17 | 72.12 | 75.13 |

**Ablation Study**  To analyze the contribution of each proposed component, we conduct an ablation study on the proposed APPLe. As shown in Table 3, we decompose the complete framework into different components: prototypes - utilizing 50 prompt prototypes to cover the large visual variance; training - fine-tuning the text features of the prompt prototypes; attention - learning an adaptive attention matrix to weight different prototypes for prediction; $\ell_{max}$ - applying an extra cross-entropy loss on the closest prototypes; $\ell_{dec}$ - applying a decorrelation loss to suppress the co-occurrence of multiple prototypes. The results confirm that each component makes its unique contribution to the complete method.

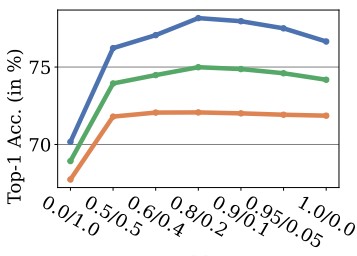

Figure 5: Performance comparison of APPLe with different mean/max calibration ratio.

**Prototype Calibration**  To investigate the contribution of averaged confidence and the maximum confidence among prototypes, we calibrate the inference confidences. As shown in Fig. 5, we choose to balance the weights of averaged logits and the maximum logits of the prototypes with 0.5/0.5, 0.6/0.4, 0.8/0.2/, 0.9/0.1, and 0.95/0.05, respectively. This calibration ratio provides insight into the contribution of inter-class and intra-class discriminative power. In APPLe, we consider both the overall prototype representations and the closest prototypes. It can be seen that when we aggregate 0.8 of the confidence from averaged logits and 0.2 of the maximum logits, we can achieve the best calibration results.

Table 4: Performance comparison on the image retrieval task with ImageNet (mAP(%))

|  | @1 | | | @5 | | | @20 | | | @50 | | |
|---|---|---|---|---|---|---|---|---|---|---|---|---|
|  | Base | New | HM | Base | New | HM | Base | New | HM | Base | New | HM |
| CLIP | **85.60** | 82.00 | 83.76 | 83.40 | 79.64 | 81.48 | 77.32 | 71.28 | 74.17 | 61.68 | 54.56 | 57.90 |
| CoOp | 82.80 | 79.80 | 81.27 | 81.12 | 77.68 | 79.37 | 75.82 | 69.67 | 72.62 | 62.14 | 53.90 | 57.73 |
| MaPLe | 84.80 | 83.40 | 84.09 | 83.28 | 81.88 | 82.57 | 77.59 | 73.43 | 75.45 | 63.23 | 56.98 | 59.94 |
| APPLe* | 85.20 | **88.20** | 86.67 | 83.76 | **84.92** | 84.34 | 77.62 | **77.30** | 77.46 | 63.01 | **61.01** | 61.99 |
| APPLe | **85.60** | 88.00 | **86.78** | **84.28** | 84.76 | **84.52** | **78.80** | 77.27 | **78.03** | **63.99** | **61.01** | **62.46** |

**Understanding Prototypes by Image Retrieval**    To understand the effect of prototypes in the inference process, we performed a series of experiments on image retrieval. In these experiments, we use the prompt prototypes to retrieve the closest image samples based on the cosine similarity. The images we aim to retrieve are from the test samples of ImageNet. There are in total 25,000 images from 1,000 categories, 50 images for each category. We cast the prompt prototypes and the images into the common space, retrieving the closest 1, 5, 20 and 50 images respectively. As shown in Table 4, APPLe can consistently retrieve more accurate images than MaPLe, CoOp and zero-shot CLIP. APPLe* represents the training-free version of our method. It is clear that APPLe* can already achieve much better performance than baseline methods. Note that Co-CoOp cannot be applied in image retrieval tasks, because the textual features need to be conditioned on images.

As for retrieving the closest images with individual prompt prototypes, there are usually many failure cases. The visual features regarding these keywords are highly salient in the wrongly retrieved images, making them closer to the prototypes. We take one step further to investigate a particular prompt, as shown in Fig. 6 We calculate the text cosine similarity between the two category names, *i.e.,* cobwebbed, and lacelike, and the keywords in the prompt. All these keywords are highly correlated with lacelike, instead of the ground-truth label cobwebbed. Therefore, we may easily retrieve a lancelike image with this prompt.

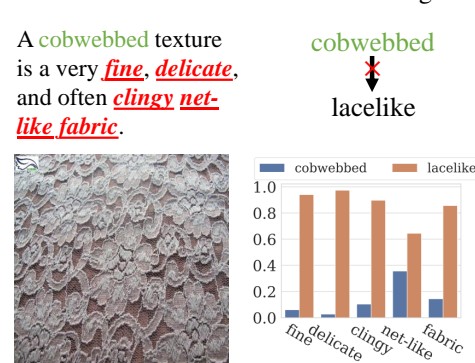

A cobwebbed texture is a very ***fine***, ***delicate***, and often ***clingy net-like fabric***.

Figure 6: Text cosine similarity between the category names and the keywords.

**Limitation of APPLe**    The primary constraints of APPLe include its necessity for fine-tuning new prototypes to optimize performance when adapting to certain new classes. Its effectiveness is significantly tethered to the quality of the prompts, as the prototypes may contain flawed keywords that can lead to ambiguous decisions, as evident from the retrieval experiments. Additionally, compared to context optimization methods, the time complexity of APPLe is less favorable due to the requirement of producing multiple prompt prototypes for each class.

## 6 CONCLUSION

In this work, we have proposed an Adaptive Prompt Prototype Learning (APPLe) method for vision-language models. By incorporating multiple prompts as class prototypes, we can largely enhance zero-shot CLIP performance. To alleviate the noise and flaws within the prompts, we designed an adaptive attention mechanism. It assigns lower confidence to the logits produced by the flawed prompts, and higher confidence to the accurate and representative prototypes. In addition, it is unlikely for a visual sample to align closely with every prototype. Thus, a prototype decorrelation loss is introduced to suppress the co-occurrence of multiple confident prototypes. APPLe demonstrates consistent performance gains on all 11 datasets and all tasks.

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
