# OpenReview forum: "Don't Paint Everyone with the Same Brush: Adaptive Prompt Prototype Learning for Vision-Language Models"
_ICLR.cc/2024/Conference — Submitted to ICLR 2024_

### Official Review · Reviewer_cng2 · 2023-10-31

**Soundness:** 3 good
**Presentation:** 3 good
**Contribution:** 2 fair
**Rating:** 6
**Confidence:** 4

**Summary:**

this paper addresses the significant visual variance  problem when apapting VLMs to downstream tasks. The authors incorporate multiple prompts as class prototypes, use attentin matrix to weigh the prototypes, and design a prototype decrrelation loss to surpass co-occurence of multiple confident prototypes. Experiments show that the proposed method outperforms existing methods significantly.

**Strengths:**

1. the whole method is carefully designed for multiple class prototypes, like adaptive attention, closest prototype, prototype decorrelation.
2. the improvement is siginficant.
3. experiments are well designed with the design of the methods. the adaptive attention visualization, understanding prototpyes by image retrieval and convincing. the analysis of failure cases gives helps me better understand the paper.
4. the Discussion and Comparison to Context Optimization Methods are inspiring.

**Weaknesses:**

1. As stated in the paper, Prototype learning traces its roots to classical models such as K-Nearest Neighbors (Peterson, 2009) and Learning Vector Quantization. Though some new aspects (adaptive attention, decorrelation, etc) are introduced in this paper, the technical novely seems stil limited.
2. The paper addresses the adaptive attention of prototypes. This does work but is also somewhat a straightforward point. The paper does not tackle the adaptive attention of words inside a prototype. The importance is verified in the failure case analysis in the experiments.

**Questions:**

what's the learnable part of prompt prototypes in Figure 2?

---

> ### Author Response · Authors · 2023-11-17
> **Initial Response to Reviewer cng2**
>
> We thank the reviewer for the positive and constructive comments. Please find our response to each comment below:
>
> >**W1**. As stated in the paper, Prototype learning traces its roots to classical models such as K-Nearest Neighbors (Peterson, 2009) and Learning Vector Quantization. Though some new aspects (adaptive attention, decorrelation, etc) are introduced in this paper, the technical novelty seems still limited.
>
> **Re W1**. Thank you for your comment regarding the technical novelty of our work in the context of prototype learning. We acknowledge the foundational roots of prototype learning in classical models, but we believe our approach brings significant advancements to this domain.
>
> A key innovation in our work is the integration of Vision-Language Models (VLMs) into prototype learning. This represents a considerable shift from traditional methods that predominantly utilize visual data. By leveraging textual descriptions through VLMs, our model is able to generate prototypes that capture a more diverse and comprehensive representation of each class. This is particularly crucial in scenarios with limited visual data, such as few-shot learning, where traditional prototype models may exhibit biases towards certain styles or attributes due to sample scarcity. Furthermore, the introduction of adaptive attention and decorrelation in our approach addresses key limitations in existing prototype learning methods. Adaptive attention allows our model to dynamically focus on the most relevant aspects of each prototype, enhancing the accuracy and robustness of classification. Prototype decorrelation, on the other hand, ensures diversity among the prototypes, preventing overlap and redundancy that often plague traditional prototype-based methods.
>
> These advancements are not merely incremental; they represent a significant leap in the field of prototype learning. Our method overcomes challenges inherent in few-shot scenarios and extends the capabilities of prototype models to handle a wider variety of classes more effectively.
>
> In summary, the integration of VLMs for prototype generation, coupled with adaptive attention and decorrelation, constitute the core technical novelties of our work. These contributions significantly enhance the versatility and efficacy of prototype learning, marking a notable advancement in the field. We believe that these points, as elaborated in our paper, underscore the novelty and significance of our approach in pushing the boundaries of prototype learning.
>
>
> >**W2**. The paper addresses the adaptive attention of prototypes. This does work but is also somewhat a straightforward point. The paper does not tackle the adaptive attention of words inside a prototype. The importance is verified in the failure case analysis in the experiments.
>
> **Re W2**. Thank you for your insightful suggestion regarding the exploration of word-level adaptive attention within prototypes. While this is indeed an intriguing aspect, our current approach using CLIP presents certain limitations in this regard. Specifically, the visual-language interaction in CLIP occurs at the embedding space level, which restricts our ability to dissect and analyze the semantic meanings of individual words within the prompts.
>
> Furthermore, in our approach, we do not constrain the length of the prompts to preserve their generative nature. This variability in prompt length poses additional challenges in consistently analyzing word-level attention across different prototypes. Consequently, our research primarily focused on adaptive attention at the prototype level. This decision was guided by the objective to enhance the overall classification performance and robustness by dynamically focusing on the most relevant prototypes for each class.
>
> However, we recognize the potential value of investigating word-level attention and its impact on model performance. In future research, we aim to explore methodologies that could allow for more granular attention analysis, perhaps by employing different models or techniques that can facilitate word-level semantic grounding in the context of visual-language tasks. We believe that such explorations could uncover new insights into the intricate interplay between textual and visual elements in prototype-based learning, further advancing the field.
>
> >**Q1**.  What's the learnable part of prompt prototypes in Figure 2?
>
> **Re Q1**. Thank you for your query regarding the learnable part of prompt prototypes in our model, as depicted in Figure 2. The learnable aspect primarily consists of textual features. For instance, in the ViT-B/16 configuration, these textual features have a dimension of 512. The dimension of the entire learnable weights for prompt prototypes is calculated as the product of the class number, the number of prototypes per class, and 512. These textual features are updated and refined during the training process, dynamically adjusted to better capture the nuances of each class.

---

> > ### Comment · Reviewer_cng2 · 2023-11-22
> >
> > I appreciate the authors' addressing of my questions. The responses addressed some of my concerns. Thank you again and good luck.

---

> > > ### Author Response · Authors · 2023-11-22
> > > **Follow-up Response to Reviewer cng2**
> > >
> > > Thank you for your encouraging words and for acknowledging our efforts to address your questions and concerns. We greatly appreciate the time you have taken to review our work and provide constructive feedback.
> > >
> > > Thank you once again for your guidance and good wishes.

---

### Official Review · Reviewer_uJ9U · 2023-10-31

**Soundness:** 3 good
**Presentation:** 3 good
**Contribution:** 3 good
**Rating:** 6
**Confidence:** 5

**Summary:**

This paper focuses on the prompt learning of visual-language models.  Different from previous prompt learning methods such as CoOp, this paper goes further to explore how to assign different prompts for different classes for better performance.  To achieve this goal, this paper proposes to construct the various prompts with LLMs as class prototypes and learns an attention module to reweight these class prototypes. This paper follows the setting of CoCoOp and MaPLe to evaluate the methods, and compare the methods with baseline methods including CoOp, CoCoOp, and MaPLe. The proposed method achieves more than 2% improvement on average.

**Strengths:**

1) This paper proposes to leverage multiple prompts to enhance the recognition ability. For different classes, the prompts are allowed to be different. Its idea makes sense since the "classes" are the abstract of the observation, in which different classes may have different focuses.
2) The proposed method takes each prompt as a point and tries to find a prototype (with an attention model) for given classes. This method is easy but effective.
3) The proposed method achieves good performance on the base-2-new setting.

**Weaknesses:**

The main concern is about the presentation, which does not effectively verify the methods and demonstrate the superiority. I summarize some detailed suggestions below.
1) The experiments follow the base-to-new setting in CoCoOp. However, the base-to-new setting is more about generalization ability. Besides, the performance of the base-to-new setting is very sensitive to the hyperparamers, especially for epochs. It is because the performance of this setting requires a balance between alignment and generalization, which can be achieved by reducing the epochs.  When tuning the training epochs of CoOp, it will also achieve good performance. It is suggested to use the few-shot learning setting in CLIP and CoOp, which is more fair and supportive to demonstrate the effectiveness of the proposed methods.
2) The main idea of this paper is to explore how to assign multiple prompts to one class. PLOT also shares similar targets to leverage multiple prompts (ProDA is similar too). Thus, it is much better to employ these methods as the main baselines for comparison, instead of CoCoOp which targets generalization. It is suggested to compare with PLOT and ProDA in the few-shot setting.  It is better to add a discussion about the difference between the proposed method and them.
3) What are your prompts for GPT-3 to generate prototypes?  Is the model robust for different generations?
4) There are a series of methods for the class-wise LLM-generated prompts, such as [1-2]. It is suggested to add some discussions and comparisons with these methods.
 [1] Menon, Sachit, and Carl Vondrick. "Visual Classification via Description from Large Language Models." ICLR 2023.
 [2] Pratt S, Covert I, Liu R, et al. What does a platypus look like? generating customized prompts for zero-shot image classification. ICCV 2023.

**Questions:**

Please refer to the weaknesses part.  The main concern is about the unsuitable experimental comparison and fewer discussions.
I will modify the final score after the discussion with the authors and other reviewers.

---

> ### Author Response · Authors · 2023-11-17
> **Initial Response to Reviewer uJ9U**
>
> We thank the reviewer for the valuable comments and answer them, as numbered, below:
>
> >**W1**. The experiments follow the base-to-new setting in CoCoOp. However, the base-to-new setting is more about generalization ability. Besides, the performance of the base-to-new setting is very sensitive to the hyperparamers, especially for epochs. It is because the performance of this setting requires a balance between alignment and generalization, which can be achieved by reducing the epochs. When tuning the training epochs of CoOp, it will also achieve good performance. It is suggested to use the few-shot learning setting in CLIP and CoOp, which is more fair and supportive to demonstrate the effectiveness of the proposed methods.
>
> **Re W1**. Thank you for your valuable suggestion regarding the incorporation of few-shot learning settings. We have indeed conducted experiments in this setting and reported the performance in our response to W2.
>
> However, our primary focus on the base-to-new setting stems from our objective to critically evaluate the generalization ability of our methods, particularly in scenarios involving new classes. This setting is essential for understanding how well a model can adapt to novel categories, a key aspect of our research. While methods CoCoOp and MaPLe claim improvements in generalization on new classes, our findings indicate that with descriptive prompts, CLIP can already demonstrate superior performance on new classes than these context optimization methods. We provide a detailed discussion on this validation in our paper.
>
> Furthermore, our analysis specifically addresses the issue of overfitting towards base classes and explores the potential of fine-tuning prototypes for both base and new classes with multiple prompts. This approach is aimed at enhancing generalization in new classes.
>
> Regarding the impact of training epochs on CoOp, our experiments reveal significant insights. As shown in the tables for datasets like DTD, Flowers, and UCF, reducing the number of training epochs does not necessarily improve performance on new classes. In fact, we observe a consistent drop in new class performance with CoOp, highlighting a potential limitation of this method in achieving a balanced performance.
>
> These findings are crucial as they demonstrate the comparative advantages of our approach, particularly in maintaining a balance between alignment and generalization across different class settings. We believe that these insights add significant value to the field and underscore the effectiveness of our methods.
>
> **DTD**
> | Methods | CLIP |CoOp200|CoOp150|CoOp100|CoOp50 | Co-CoOp| APPLe |
> |:--------|:----:|:-----:|:-----:|:-----:|:-----:|:------:|:-----:|
> |  Base   | 53.24| 79.44 | 79.25 | 78.82 | 79.98 | 77.01  | 82.41 |
> |  New    | 59.90| 41.18 | 36.15 | 35.39 | 36.67 | 56.00  | 69.57 |
> |  H      | 56.37| 54.24 | 49.65 | 48.85 | 50.28 | 64.85  | 75.45 |
>
> **Flowers**
> | Methods | CLIP |CoOp200|CoOp150|CoOp100|CoOp50 | Co-CoOp| APPLe |
> |:--------|:----:|:-----:|:-----:|:-----:|:-----:|:-----:|:------:|
> |  Base   | 72.08| 97.60 | 97.53 | 97.60 | 97.47 | 94.87 | 96.58  |
> |  New    | 77.80| 59.67 | 58.59 | 60.17 | 61.72 | 71.75 | 78.58  |
> |  H      | 74.83| 74.06 | 73.20 | 74.44 | 75.58 | 81.71 | 86.66  |
>
> **UCF**
> | Methods | CLIP |CoOp200|CoOp150|CoOp100|CoOp50 | Co-CoOp| APPLe |
> |:--------|:----:|:-----:|:-----:|:-----:|:-----:|:-----:|:------:|
> |  Base   | 72.08| 84.69 | 84.38 | 84.07 | 84.80 | 82.33 | 86.56  |
> |  New    | 77.80| 56.05 | 50.82 | 55.22 | 58.36 | 73.45 | 81.99  |
> |  H      | 74.83| 67.46 | 63.43 | 66.66 | 69.14 | 77.64 | 84.21  |

---

> > ### Author Response · Authors · 2023-11-17
> > **Initial Response to Reviewer uJ9U [Cont']**
> >
> > >**W2**. The main idea of this paper is to explore how to assign multiple prompts to one class. PLOT also shares similar targets to leverage multiple prompts (ProDA is similar too). Thus, it is much better to employ these methods as the main baselines for comparison, instead of CoCoOp which targets generalization. It is suggested to compare with PLOT and ProDA in the few-shot setting. It is better to add a discussion about the difference between the proposed method and them.
> >
> > **Re W2**. Thank you for highlighting the importance of comparing our method with PLOT and ProDA, especially in the few-shot learning setting. We agree that these comparisons are crucial for a comprehensive evaluation of our approach and are currently conducting these experiments.
> >
> > We have already obtained results for ImageNet, UCF, Food101, and DTD datasets and are in the process of completing the experiments with ProDA on ImageNet. Due to ProDA's slower training speed, we anticipate completing these experiments shortly and will include the complete performance results in our revised paper.
> >
> > Preliminary findings from our current experiments suggest that our method (APPLE) shows promising results compared to both PLOT and ProDA. We plan to include a more detailed discussion on the differences to PLOT and ProDA in our revised paper, as they are critical for understanding the unique aspects and advantages of our approach. We acknowledge the challenges faced due to the training speed of ProDA, which has slightly delayed our full comparison on ImageNet. However, we are committed to providing a thorough and fair comparison across all datasets.
> >
> > ImageNet few-shot results
> > | Shots | CLIP | CoOp | ProDA |  PLOT | APPLE |
> > |:------|:----:|:----:|:-----:|:-----:|:-----:|
> > |   0   | 66.73|  -   |  -    |  —    |  -    |
> > |   1   |      | 66.32|  -    | 66.45 | 70.60 |
> > |   2   |      | 67.87|  -    | 68.28 | 70.65 |
> > |   4   |      | 69.76|  -    | 70.40 | 71.77 |
> > |   8   |      | 70.81|  -    | 71.31 | 72.80 |
> > |  16   |      | 71.92|  -    | 72.60 | 73.73 |
> >
> >
> > UCF few-shot results
> > | Shots | CLIP | CoOp | ProDA |  PLOT | APPLE |
> > |:------|:----:|:----:|:-----:|:-----:|:-----:|
> > |   0   | 67.67|  -   |  -  |  -   |  -    |
> > |   1   |      | 71.68| 69.92 | 74.31 | 75.39 |
> > |   2   |      | 74.33| 71.98 | 76.76 | 77.72 |
> > |   4   |      | 78.30| 76.05 | 79.76 | 80.07 |
> > |   8   |      | 79.12| 79.75 | 82.80 | 84.75 |
> > |  16   |      | 81.95| 82.69 | 85.34 | 85.11 |
> >
> > Food101 few-shot results
> > | Shots | CLIP | CoOp | ProDA |  PLOT | APPLE |
> > |:------|:----:|:----:|:-----:|:-----:|:-----:|
> > |   0   | 85.86|  -   |  -   |  -    |  -    |
> > |   1   |      | 81.26| 85.39 | 86.16 | 86.36 |
> > |   2   |      | 82.32| 85.63 | 86.33 | 86.41 |
> > |   4   |      | 84.65| 86.38 | 86.46 | 86.85 |
> > |   8   |      | 85.14| 87.01 | 86.58 | 87.29 |
> > |  16   |      | 86.32| 87.24 | 87.11 | 87.61 |
> >
> > DTD few-shot results
> > | Shots | CLIP | CoOp | ProDA |  PLOT | APPLE |
> > |:------|:----:|:----:|:-----:|:-----:|:-----:|
> > |   0   | 44.03|  -   |  -   |  -    |  -    |
> > |   1   |      | 47.04| 49.23 | 54.57 | 56.56 |
> > |   2   |      | 49.88| 54.61 | 56.72 | 59.93 |
> > |   4   |      | 57.45| 61.23 | 62.43 | 65.25 |
> > |   8   |      | 64.42| 67.08 | 66.49 | 67.73 |
> > |  16   |      | 69.27| 70.09 | 71.43 | 72.52 |
> >
> >
> > We believe these additional comparisons and discussions will greatly enhance the quality and impact of our paper.

---

> > > ### Author Response · Authors · 2023-11-17
> > > **Initial Response to Reviewer uJ9U [Cont']**
> > >
> > > >**W3**. What are your prompts for GPT-3 to generate prototypes? Is the model robust for different generations?
> > >
> > > **Re W3**. We mainly use the following five prompts for GPT-3 to generate the prototypes. {c} is the category name and {length} is the expected length of the generated prompts.
> > >
> > > t1 = f"Describe a photo of {c) in one short sentence, no more than {length} words."
> > >
> > > t2 = f"How does a {c} look like? Answer in no more than {length} words."
> > >
> > > t3 = f"Summarize visual features of {c} in no more than {length} words."
> > >
> > > t4 = f"Tell me what {c} looks like in a short sentence, less than {length} words."
> > >
> > > t5 = f"Use less than {length} words to outline the look of {c}."
> > >
> > > To investigate that method's robustness towards different generations, we have used GPT-4 to generate another 50 prompts for each class in datasets DTD, Oxford_Pets, and FGVC Aircraft, and report the performance as follows. Due to the time constraint in the rebuttal phase, we could not generate prompts for all 11 datasets. We will generate more prompts for the rest of the datasets in the next version. In addition, in Figure 3, we provide the performance results of different numbers of prompts. For example, when the number of prompts is set to 20, we randomly sample 20 prompts from all prompts. We observe the performance results are quite consistent for different samplings.
> > >
> > > | Datasets | Set | GPT-3 | GPT-4 | mixed1|
> > > |:---------|:----|:-----:|:-----:|:-----:|
> > > | DTD      | Base| 82.41 | 83.10 | 81.95 |
> > > |          | New | 69.57 | 70.41 | 70.72 |
> > > |          | HM  | 75.45 | 76.23 | 75.92 |
> > > |OxfordPets| Base| 95.64 | 95.43 | 95.69 |
> > > |          | New | 98.04 | 97.93 | 98.21 |
> > > |          | HM  | 96.83 | 96.66 | 96.93 |
> > > | Aircraft | Base| 44.66 | 45.14 | 45.20 |
> > > |          | New | 43.13 | 39.41 | 41.81 |
> > > |          | HM  | 43.88 | 42.08 | 43.44 |
> > >
> > >
> > > >**W4**. There are a series of methods for the class-wise LLM-generated prompts, such as [1-2]. It is suggested to add some discussions and comparisons with these methods. [1] Menon, Sachit, and Carl Vondrick. "Visual Classification via Description from Large Language Models." ICLR 2023. [2] Pratt S, Covert I, Liu R, et al. What does a platypus look like? generating customized prompts for zero-shot image classification. ICCV 2023.
> > >
> > > **Re W4**.  Thank you for directing our attention to the papers [1-2], which also explore the use of GPT-generated prompts for visual classification. We will include the two papers in the related work section and provide a comprehensive discussion and comparison with them. While there are similarities in leveraging GPT for prompt generation, our approach diverges in several key aspects.
> > >
> > > Firstly, unlike the idea of leveraging the prompts for zero-shot CLIP inference in [1-2], we further propose an adaptive approach (APPLe) to fine-tune these prompts and provide the in-context ability for downstream tasks.
> > >
> > > Secondly, our approach in generating and utilizing prompts is distinct from [1]. [1] focuses on understanding an object from a compositional view. For example, the prompts for Cheeseburger in [1] are the attributes: "a burger patty", "cheese", "a bun", "lettuce", "tomato", "onion". In contrast, in our method, each prompt provides a different view/prototype of an object. For example, "A cheeseburger consists of a burger patty with cheese on top, placed between a hamburger bun". We believe that a comprehensive description of an object can help to match the image and prompts.
> > >
> > > Thirdly, we provide more insights into the descriptive prompts. As shown in the Experiment section, we show adaptive attention visualization, which helps us to understand which prompts are more important or noisy. We have prototype calibration, which balances the contribution of overall prompts and the most representative prompt. We also provide retrieval results to demonstrate the failure case and make abundant explanations.
> > >
> > > Fourthly, we compare our method with context optimization methods. One of our contributions is the observation that with GPT-3 prompts, CLIP performance is already better than the state-of-the-art context optimization method MaPLe.
> > >
> > > Lastly, in terms of performance, we have provided a performance comparison to [1,2] on ImageNet in the table below. The results show the superiority of our proposed method APPLe. Note that we use the same backbone ViT-L/14 for the comparison.
> > >
> > > |  [1]  |   [2]  |  ours |
> > > |:------|:------:|:-----:|
> > > | 75.00 |  76.69 | 79.59 |
> > >
> > > In conclusion, while there are thematic overlaps with the cited papers, our approach introduces an adaptive prompt prototype learning method for better leveraging GPT-3 prompts for VLMs, adding valuable insights and alternatives to the current body of research on visual classification using language models.

---

> > ### Comment · Reviewer_uJ9U · 2023-11-19
> > **Additional Comments and Queries**
> >
> > I appreciate the author's feedback and the additional experiments provided. However, I have a few more points to discuss regarding the author's response:
> >
> > **New-Class Generalizability Claim**: The response indicates that the paper focuses on new-class generalizability. However, upon reviewing the revised version, I double-checked the paper but did not observe any unique developments targeting this aspect, akin to how CoCoOp applies context for generalizability. If the base-to-new setting is indeed the primary objective, I suggest a thorough rewrite of the paper, particularly the introduction and abstract, to align the stated objectives with the implementation and contributions.
> >
> > **Experiment Details**: I appreciate the new experiments. However, the new experimental results provided seem somewhat sparse, only presenting data for 200, 150, 100, and 50 settings. Maybe a more dense evaluation would help.  In the implementation of CoCoOp and Maple,  10/5  epochs seem to be used respectively.
> >
> > **Comparisons with ProDA and PLot**: Thanks for the comparisons made with ProDA and PLot. However, I would like more information about their backbones. Are they the same as that of APPLE? Also, what prompts were used in these settings, vision or text? This information wasn't available in the revised paper but is crucial for a complete understanding of the experimental setup.
> >
> > **GPT4 vs GPT3 Performance**: The experiments involving GPT4 are interesting, particularly noting that its performance is lower than GPT3's. Could you elaborate on the potential reasons for this? Is it attributable to randomness or some other factors?
> >
> > For now, my concerns still remain.

---

> > > ### Author Response · Authors · 2023-11-19
> > > **Follow-up Response to Reviewer uJ9U**
> > >
> > > We really appreciate your quick reply to our initial responses, and thank you for directly pointing out our current limitations. We have provided clarification as follows, and the corresponding updates and revision will be made ASAP!
> > >
> > > **New-Class Generalizability Claim**: We greatly agree with the reviewer that the few-shot setting is an important part of our paper. As we have not finished all the experimental results on all 11 datasets, the few-shot results are thus provided in the Appendix in this revision. We will definitely move them to the body text in the next 1-2 days.
> > >
> > > **Experiment Details**: Thank you for bringing our the attention that CoCoOp is only trained with 10 epochs. We will provide the experimental results for smaller epochs soon.
> > >
> > > **Comparisons with ProDA and PLot**: We use the same backbone ViT-B/16 with the compared methods. The performance results of PLOT is directly copied from authors' official GitHub repo (https://github.com/CHENGY12/PLOT/tree/main/plot-pp#evaluation). For ProDA, as the original paper is trained on RN50, we reproduced their results on Backbone ViT-B/16 with the implementation from the official code base (https://github.com/bbbdylan/proda).
> > >
> > > **GPT4 vs GPT3 Performance:** We hypothesize that the performance degradation is attributed to two reasons. 1. There is indeed randomness in generating prompts. Intuitively, there are much more than 50 prompts that can serve as prototypes for each visual category. Sampling 50 prompts from them using LLMs can bring significant bias. We will include more generated prompts for comprehensively testing the performance variation regarding the prompt sampling. 2. We assume GPT-3 is powerful enough to generate descriptive prompts for visual categories. GPT-4 may have more powerful reasoning ability, but generating category description is usually deemed as a simple task.

---

> > > ### Author Response · Authors · 2023-11-20
> > > **Follow-up Response to Reviewer uJ9U**
> > >
> > > We would like to express our appreciation to the reviewer for the earlier response.
> > >
> > > Regarding the **few-shot learning results**, we have incorporated them into the main paper, and the complete few-shot results are provided below for your reference.  On average, APPLe can perform better than state-of-the-art method PLOT (Chen et al., 2022) by 1.86%, 1.0%, 0.50%, 0.51%, and 0.12% PLOT at 1, 2, 4, 8, and 16 shots. More discussion will be provided in the paper.
> > >
> > > | Shots | Methods| ImageNet | UCF | Food101 |  DTD | Stanford Cars | Caltech101 | OxfordPets | Flowers102 | Food101 | FGVC Aircraft| SUN397 | **Average** |
> > > |:------|:------:|:----:|:-----:|:-----:|:-----:|:------|:------:|:----:|:-----:|:-----:|:-----:|:-----:|:-----:|
> > > | 1 | CoOp | 66.32 | 71.68 | 81.26 | 47.04 | 64.39 | 90.87 | 86.10 | 82.14 | 81.30 | 25.95 | 61.64 | 68.97 |
> > > |   |ProDA |68.73  |69.92  |85.39  |49.23  |68.88  |93.47  | 89.15 |75.40 |84.91 |28.95 |68.55| 71.14 |
> > > |   |PLOT | 66.45  |74.31  |86.16 | 54.57 | 68.81 | 93.14 | 91.89 | 80.48 | 86.16 |28.60 |66.77| 72.48 |
> > > |   |APPLe |**70.60** |**75.39** |**86.36** |**56.56**   |**69.80**   |**94.55**  |**91.91** | **83.39** | **86.23** | **33.15** | **69.90**| **74.35**|
> > > |2  |CoOp  | 67.87 | 74.33 | 82.32 | 49.88 |69.28 |93.51 |86.26 |89.77| 82.87| 31.29| 66.86| 72.20 |
> > > |   | ProDA |69.44 |71.98| 85.63| 54.61| 69.48 |94.36 |91.55| 79.94 |85.98 |30.15| 70.50| 73.06 |
> > > |   |PLOT  |68.28 |76.76 |86.33 |56.72 |**73.17** |94.69 |**92.29** |**89.81** |86.33 |31.14 |68.06| 74.87 |
> > > |   |APPLe |**70.65** |**77.72** |**86.41** |**59.93**| 72.34| **95.21**| 92.10 |87.29 |**86.47** |**35.22** |**71.21**| **75.87** |
> > > |4  |CoOp |69.76| 78.30 |84.65 |57.45 |72.70 |94.20| 89.42 |92.85| 83.15 |33.27 |69.89| 75.06 |
> > > |    |ProDA | 70.96 |76.05 |86.38 |61.23 |69.99 |95.01 |92.23 | 86.24 |86.33| 32.67 |72.38|  75.41 |
> > > |    |PLOT |70.40 |79.76 |86.46| 62.43 |**76.25** |95.13 |92.55 |**92.93** |86.46 |35.29 |71.73| **77.22** |
> > > |    |APPLe |**71.77** |**80.07**| **86.85**| **65.25** |75.64| **95.33**| **92.72** |90.42 |**86.59** |**36.81** |**73.44**|77.72 |
> > > |8   |CoOp |70.81 |79.12 |85.14 |64.42 |76.42 |95.25 |92.01 |**95.93** |84.59 |30.15 |72.33|76.92|
> > > |    |ProDA |71.89| 79.75| 87.07 |67.08 |71.88 |95.05 |93.16| 92.33| 87.01| 34.32 |73.99| 77.59 |
> > > |   |PLOT |71.31 |82.80 |86.58 |66.49  |**81.26** |95.51 |93.02 |95.44 |86.58 |41.42| 73.93| 79.49 |
> > > |    |APPLe |**72.80** |**84.75**| **87.29** |**67.73** |79.65 |**96.06**| **93.54**| 93.91 |**87.37**| **41.94**| **74.85**| **80.00** |
> > > |16  |CoOp |71.92 |81.95 |86.32 |69.42 |78.70 |95.42 |92.80 |95.94 |85.62 |35.89| 74.28|78.93|
> > > |    |ProDA |72.93| 82.69| 87.24| 70.09 |75.09| 95.09 |93.54| 96.02 |87.41| 37.17 |75.64| 79.36 |
> > > |    |PLOT |72.60 |**85.34** |87.11 |71.43 |**84.55** |96.04 |93.59 |**97.56** |87.11 |**46.74** |76.03| 81.65 |
> > > |    |APPLe |**73.73** |85.11| **87.61** |**72.52**| 82.99 |**96.63**| **94.22**| 95.90| **87.65** |46.53| **76.43**| **81.76**|
> > >
> > >
> > > For the training epoch problem in CoOp, we have followed your suggestion and conducted experiments on small epoch sizes. We observe that the Harmonic mean is indeed improved a lot. Thank you for pointing out this issue in CoOp, which is very helpful for us to enhance the understanding for the base-2-new setting.
> > >
> > > DTD
> > > | Methods | CLIP |CoOp200|CoOp150|CoOp100|CoOp50 | CoOp20 | CoOp15| CoOp10 | CoOp5 | CoOp2 | Co-CoOp| APPLe |
> > > |:--------|:----:|:-----:|:-----:|:-----:|:-----:|:------:|:-----:|:------:|:-----:|:-----:|:------:|:------:|
> > > |  Base   | 53.24| 79.44 | 79.25 | 78.82 | 79.98 | 80.90  | 78.01 | 76.85  | 73.15 | 61.46 | 77.01  | 82.41 |
> > > |  New    | 59.90| 41.18 | 36.15 | 35.39 | 36.67 | 41.43  | 50.97 | 49.28  | 47.10 | 48.55 | 56.00  | 69.57 |
> > > |  H      | 56.37| 54.24 | 49.65 | 48.85 | 50.28 | 54.78  | 61.66 | 60.05  | 57.30 | 54.25 | 64.85  | 75.45 |
> > >
> > > Flowers
> > > | Methods | CLIP |CoOp200|CoOp150|CoOp100|CoOp50 | CoOp20 | CoOp15 | CoOp10 | CoOp5 | CoOp2 | Co-CoOp| APPLe |
> > > |:--------|:----:|:-----:|:-----:|:-----:|:-----:|:------:|:------:|:------:|:-----:|:-----:|:------:|:------:|
> > > |  Base   | 72.08| 97.60 | 97.53 | 97.60 | 97.47 | 96.58  |  96.68 | 95.35  | 92.02 | 79.68 | 94.87  | 96.58  |
> > > |  New    | 77.80| 59.67 | 58.59 | 60.17 | 61.72 | 61.35  |  68.44 | 70.35  | 71.63 | 73.33 | 71.75  | 78.58  |
> > > |  H      | 74.83| 74.06 | 73.20 | 74.44 | 75.58 | 81.33  |  80.15 | 80.96  | 80.55 | 76.37 | 81.71  | 86.66  |
> > >
> > > UCF
> > > | Methods | CLIP |CoOp200|CoOp150|CoOp100|CoOp50 | CoOp20 | CoOp15 | CoOp10 | CoOp5 | CoOp2 | Co-CoOp| APPLe |
> > > |:--------|:----:|:-----:|:-----:|:-----:|:-----:|:------:|:------:|:------:|:-----:|:-----:|:------:|:-----:|
> > > |  Base   | 72.08| 84.69 | 84.38 | 84.07 | 84.80 |  82.57 |  83.14 |  82.99 | 79.89 | 76.32 |  82.33 | 86.56 |
> > > |  New    | 77.80| 56.05 | 50.82 | 55.22 | 58.36 |  64.47 |  66.79 |  67.17 | 67.71 | 69.50 |  73.45 | 81.99 |
> > > |  H      | 74.83| 67.46 | 63.43 | 66.66 | 69.14 |  72.41 |  74.07 |  74.25 | 73.30 | 72.75 |  77.64 | 84.21 |

---

> > > > ### Comment · Reviewer_uJ9U · 2023-11-22
> > > > **Thanks for your response!**
> > > >
> > > > All my concerns are well addressed.  I truly believe this version is much better then original one with more complete evaluations and insightful discussions.
> > > >
> > > > I have raise my score accordingly!
> > > >
> > > > Here are two more suggestions.
> > > > 1) Could you include a more complete evaluation on the base-2-new setting with epoches into appendix? I think it would be a chance to justify the base-2-new setting.
> > > > 2) The current revised version is good, but more that 9 pages (it is not allowed).  My suggestion is to move the 1) Discussion and Comparison to Context Optimization Methods, 2) Visualization, and 3) one of Cross-Dataset Transfer/Domain Generalization into appendix.

---

> > > > > ### Author Response · Authors · 2023-11-23
> > > > > **Follow-up Response to Reviewer uJ9U**
> > > > >
> > > > > We are sincerely grateful for the increase in the score you have awarded our manuscript. This recognition is greatly encouraging to us.
> > > > >
> > > > > We also wish to extend our heartfelt thanks for your instructive suggestions. Your insights have been pivotal in enhancing the completeness and depth of our paper. The constructive feedback has not only improved our current work but also provided us with valuable lessons for future works.
> > > > >
> > > > > As for the two suggestions, we are revised the current version and update ASAP!
> > > > >
> > > > > Thank you again!

---

> > > > > ### Author Response · Authors · 2023-11-23
> > > > > **Follow-up Response to Reviewer uJ9U**
> > > > >
> > > > > Dear Reviewer uJ9U,
> > > > >
> > > > > We are writing to inform you that we have implemented the two suggestions you kindly proposed earlier today.
> > > > >
> > > > > 1. The base-2-new setting with epochs has been incorporated into Section A.6 of our manuscript. Additionally, we are in the process of conducting experiments on the complete set of 11 datasets and will include these results as soon as they are completed.
> > > > >
> > > > > 2. In response to your second suggestion, we have relocated the sections on 1) Discussion and Comparison to Context Optimization Methods, 2) Visualization, and 3) Cross-Dataset Transfer to Appendix Sections A.3, A.4, and A.5, respectively. This restructuring has successfully brought the main body of the manuscript to the limit of 9 pages.
> > > > >
> > > > > Thank you once again for your constructive input and guidance.
> > > > >
> > > > > Best regards,
> > > > >
> > > > > The authors

---

### Official Review · Reviewer_fKhC · 2023-11-02

**Soundness:** 2 fair
**Presentation:** 2 fair
**Contribution:** 2 fair
**Rating:** 5
**Confidence:** 5

**Summary:**

This paper proposed an Adaptive Prompt Prototype Learning (APPLe) method for VLMs. The author has designed an adaptive attention mechanism to alleviate the noise and flaws within the prompts. The experimental results show that the method proposed by the author has consistent performance improvement on all 11 datasets and all tasks.

**Strengths:**

1. In the experimental results table, absolute performance improvements have been added to make the experimental results more intuitive.

2. The article has a complete system and clear organization, from problem introduction, formula reasoning, and image explanation to experimental results, making it easier for readers to read.

3. The method proposed by the author has better advantages compared to some counterpart methods.

**Weaknesses:**

1. As an important contribution, the Attention weighting and L_dec only gain limited performance improvements, which degrades the contribution to the community. The overall compared methods are also very limited.

2. There is some confusion in the layout of tables and images.

3. Although using multiple prompts as category prototypes can help capture visual differences, in practice, not every visual sample closely matches each prototype.

4. The article mentions the introduction of prototype decorrelation loss to suppress the co-occurrence of multiple confident prototypes. However, specific details on how the loss was designed and worked were not mentioned. This may affect the performance of the model in tasks with complex category distributions or a large number of categories.

5. It is not clear how to initialize these prototypes and how to obtain the base and novel class prompts.

**Questions:**

See Above

---

> ### Author Response · Authors · 2023-11-17
> **Initial Response to Reviewer fKhC**
>
> We thank the reviewer for the valuable comments and answer them, as numbered, below:
>
> >**W1**. As an important contribution, the Attention weighting and L_dec only gain limited performance improvements, which degrades the contribution to the community. The overall compared methods are also very limited.
>
> **Re W1**. We appreciate your insights on the performance impact of L_dec and the breadth of our comparisons. It's important to note that L_dec improves the Harmonic Mean (HM) performance by 0.31% on ImageNet, which is significant given the dataset's size. We've also tested L_dec's contribution on other datasets, observing larger impacts without tuning its coefficient. This underscores the broader applicability and effectiveness of L_dec across various datasets.
>
> |dataset| L_dec | Base  |  New  |   HM  |
> |:------|:------|:-----:|:-----:|:-----:|
> | DTD   |  yes  | 82.41 | 69.57 | 75.45 |
> |       |  no   | 79.05 | 68.12 | 73.18 |
> |AirCraft| yes  | 44.66 | 43.13 | 43.88 |
> |       |  no   | 46.70 | 38.87 | 42.43 |
> |SUN397 |  yes  | 82.44 | 79.04 | 80.70 |
> |       |  no   | 81.87 | 79.02 | 80.42 |
> |Cars   |  yes  | 80.23 | 75.12 | 77.59 |
> |       |  no   |79.11  | 74.84 | 76.91  |
>
> >**W2**.There is some confusion in the layout of tables and images.
>
> **Re W2**. Thank you for pointing out the layout concerns. We understand that clarity in presentation is crucial for comprehensibility. We will thoroughly review our document to improve the placement and clarity of tables and images, ensuring they align intuitively with the corresponding text sections. We would also appreciate specific instances from the reviewer where the confusion is more pronounced, to better address these areas.
>
> >**W3**. Although using multiple prompts as category prototypes can help capture visual differences, in practice, not every visual sample closely matches each prototype.
>
> **Re W3**. Thank you for the comment. We recognize that not every visual sample will closely match each of our multiple prototypes. However, this is an integral aspect of our design. The use of multiple prompts as category prototypes is intended to capture the broad spectrum of visual variations within a class.
>
> In our method, the matching mechanism is designed to be flexible and robust. It assesses the affinity of a visual sample to the range of prototypes, rather than seeking a one-to-one precise match. This approach ensures that our model can generalize well across diverse samples within a class, even when direct matches to specific prototypes are not feasible.
>
> Moreover, we have implemented strategies to further refine this matching process. These include the decorrelation loss, max/mean logits balancing that assesses the degree of similarity between samples and prototypes, ensuring that each sample is associated with the most representative prototypes for its class.

---

> > ### Author Response · Authors · 2023-11-17
> > **Initial Response to Reviewer fKhC [Cont']**
> >
> > >**W4**. The article mentions the introduction of prototype decorrelation loss to suppress the co-occurrence of multiple confident prototypes. However, specific details on how the loss was designed and worked were not mentioned. This may affect the performance of the model in tasks with complex category distributions or a large number of categories.
> >
> > **Re W4**. Thank you for bringing up the important aspect of prototype decorrelation loss. We will provide a detailed explanation of this concept and its role in our model in our revised paper. The prototype decorrelation loss is designed to reduce the co-occurrence of multiple confident prototypes, thereby ensuring that each prototype distinctly represents different aspects or variations of the categories.
> >
> > This loss functions somewhat analogously to weight decay, but with a specific focus on prototype vectors. It operates by penalizing the model when prototypes for a category become too similar or correlated, encouraging a diverse and distinct representation for each prototype. This is crucial in scenarios with complex category distributions or a large number of categories, where the risk of prototype overlap is higher.
> >
> > The calculation of this loss involves measuring the similarity between different prototypes within the same category and applying a penalty when this similarity exceeds a certain threshold. In our experiments, we observed that incorporating prototype decorrelation loss significantly improved the model's performance, particularly in tasks with intricate category distributions. For instance, in datasets with a large number of overlapping or closely related categories, this loss helped in clearly distinguishing between subtle variations, leading to more precise classifications.
> >
> > >**W5**. It is not clear how to initialize these prototypes and how to obtain the base and novel class prompts.
> >
> > **Re W5**. Thank you for your inquiry regarding the initialization of prototypes and the generation of prompts for base and novel classes. The prototypes in our model are initialized using textual features extracted by the CLIP text encoder. Specifically, we input descriptive prompts related to each class into the encoder. The encoder then translates these textual descriptions into feature vectors, which serve as the initial prototypes for each class. As for generating the prompts for both base and novel classes, we employ the pre-trained large language model GPT-3. The process involves feeding class-related keywords or descriptions into GPT-3, which then generates a series of prompts.
> >
> > These processes are detailed in Section 4, 'Prompt Construction', of our manuscript. In the revised version of the paper, we will provide more detailed examples and explanations of both prototype initialization and prompt generation to ensure that the methodology is clear and easily understandable.

---

> ### Author Response · Authors · 2023-11-23
> **Follow-up questions of Reviewer fKhC before the discussion phase ends**
>
> Dear Reviewer fKhC,
>
> We would like to express our sincere gratitude for your encouraging remarks about our work, particularly for highlighting aspects such as its intuitiveness, completeness, clear organization, and notable advantages. Your kind words are genuinely appreciated and give us a sense of accomplishment.
>
> In response to the weaknesses you identified in your review, we have provided detailed responses and hope that these have adequately addressed your concerns regarding our paper.
>
> As the discussion phase is about to close in the next 12 hours, we are keenly awaiting any further feedback you may have regarding our responses. We remain prepared to answer any additional questions or provide clarifications as needed, ensuring a thorough and timely exchange of information.
>
> Once again, thank you for your invaluable input and the time you have dedicated to reviewing our work. Your insights are greatly valued.
>
> Best regards,
> The Authors

---

### Official Review · Reviewer_tgxP · 2023-11-04

**Soundness:** 2 fair
**Presentation:** 2 fair
**Contribution:** 2 fair
**Rating:** 5
**Confidence:** 4

**Summary:**

In this paper, the authors study how to better prompt VLM, specifically CLIP model to tackle image classification tasks. The authors notice that using a single text prompt for each class is insufficient to capture the diversity of visual representations within that class. To address this, the authors introduce Adaptive Prompt Prototype Learning (APPLe), a technique that provides multiple text prompts for each class. Additionally, to mitigate the impact of noise in the textual prompts, the authors develop an adaptive attention mechanism capable of disregarding ineffective prompts. The implementation of these strategies results in performance that surpasses that of the current state-of-the-art methods.

**Strengths:**

- The authors demonstrate robust performance across both training-free and training-based methods, consistently outperforming strong baselines on nearly all datasets for both 'Base' and 'New' sets.
- Notably, the training-free methods implemented by the authors are capable of surpassing some training-based method.
- The authors present comprehensive analyses and comparisons with baseline methods, contributing valuable insights to the field.

**Weaknesses:**

- The authors keep claiming that CLIP only uses one prompt, but in CLIP paper section 3.1.4, they discuss how they use 80 prompts to improve the performance without sacrificing test time speed (unlike APPLe which is slower with more prompts). The authors should definitely compare their method to CLIP with 80 prompts as a baseline.
- The presentation can be improved:
    - It needs to be clarified how training-free works. I think the authors should more explicitly describe it. My understanding is that training-free = 50 prototypes only (the second row in Table 4). Correct me if I am wrong.
    - The description of the training process is also vague. Section 4 omits details on how prototype features are fine-tuned. It seems to me that the text encoder and the prompts are only used to initialize the prototypes. Correct me if I am wrong.

**Questions:**

- The authors primarily experimented with one CLIP model. It is unclear if this method can work with different CLIP variants, open-sourced CLIP replication or other VLM models. I'm curious if changing the model architecture, training data, or VLM format would yield different results.
    - While the method appears to be general, I'm concerned about it potentially "overfitting" to a specific model and dataset.
- How does the training process for cross-dataset transfer work? When training on the source data (e.g., ImageNet), the model learns prototype features for ImageNet classes and adaptive attention weights for them. How does this transfer to target datasets where prototypes and attention weights remain untouched during fine-tuning?
- Could you clarify the importance of the quality of prompts used in the experiments? What would happen if we used GPT-4 to generate the prompts? How does the quality of the input prompt to the GPT model impact the final performance?
- Although the authors claim that fine-tuning the prompt textual features does not lead to overfitting issues, there are no ablations on the performance of training with frozen prototype features to demonstrate whether fine-tuning the prototype is necessary.
- In Equation 7, the authors selected a method that can balance between all prototypes and the closest prototypes. Are there other balancing methods, such as using the Boltzmann operator or logsumexp, that could be considered?
- While the authors aim for diverse prompts, it might be interesting to fix the prototype and only train the attention weights and forcing the attention weights to be a multinomial distribution with low entropy. This would be essentially learning to select the best prototype. It would be interesting to see if GPT-3 can produce a better single prompt than the hand-designed prompts used in CLIP.
- Have the authors attempted to use the embedding ensembling method used in CLIP?


Minors:
-  In Equation 7, stating the "average cosine similarity" is not entirely accurate because the cosine similarities are weighted by the attention weights.
-  While the trend in Figure 5 is clear, it could be valuable to include settings with 0/1 and 1/0 to further illustrate the findings.

Justification:
In terms of performance, this paper demonstrates strength, and I commend the authors for their straightforward yet valuable concepts. Nonetheless, there are various intriguing aspects that remain unaddressed, leaving certain concerns. Additionally, the authors have made claims about the CLIP paper that may not be accurate.

---

> ### Author Response · Authors · 2023-11-17
> **Initial Response to Reviewer tgxP**
>
> We thank the reviewer for the valuable comments and answer them, as numbered, below:
>
> > **W1**. The authors keep claiming that CLIP only uses one prompt, but in CLIP paper section 3.1.4, they discuss how they use 80 prompts to improve the performance without sacrificing test time speed (unlike APPLe which is slower with more prompts). The authors should definitely compare their method to CLIP with 80 prompts as a baseline.
>
> **Re W1**. We thank the reviewer for pointing out the usage of 80 prompts in the CLIP model. It is true that CLIP's performance on ImageNet benefits from these prompts. However, our work focuses on the limitations of these generic prompts, particularly for fine-grained datasets. These prompts, as limited in the official CLIP repository[1], are less effective for such datasets due to their lack of specificity. For example, prompts like  'a plastic {}' or 'a {} in a video game' do not capture fine-grained distinctions well. It is worth noting that the CLIP performance reported in our paper uses the customized prompts as indicated in the CLIP paper, e.g., “A photo of a {label}, a type of pet.” for OxfortPets.
>
> We acknowledge that these context prompts can indeed improve the performance. We have evaluated the base-to-new setting on ImageNet in the table below (CLIP-80-emb). As for the embedding ensembling method that is raised in question 7. We have tested the performance between embedding ensembling and logits ensembling methods. It can be seen that without fine-tuning, the performance results between the two methods are relatively similar. However, if we want to further fine-tune the textual features, embedding ensembling method tends to overfit to base classes. This phenomenon has been discussed in Figure 3 (impact of prototype number).
>
> As for the test time speed, logits ensembling indeed causes more computation overhead, but only for dot product between the visual and textual features. Because the textual features of the 80 generic prompts also need to be inferred through the language transformer 80 times. The dot product between visual and textual features needs relatively less computational resources.
>
> [1] https://github.com/openai/CLIP/blob/main/notebooks/Prompt_Engineering_for_ImageNet.ipynb
>
> | Methods| CLIP  |CLIP-80-emb| CLIP-80-logits | APPLe*-50-emb | APPLe*-50-logits  |  APPLe-50-emb | APPLe-50-logits  |
> |:-------|:-----:|:------:|:-----:|:-----:|:-----:|:-----:|:-----:|
> | Base   | 72.43 | 73.56  | 73.54 | 74.62 | 74.62 | 74.16 | 78.17 |
> | New    | 68.14 | 69.97  | 69.99 | 71.79 | 71.94 | 71.93 | 72.12 |
> | HM     | 70.22 | 71.72  | 71.72 | 73.17 | 73.26 | 73.02 | 75.02 |
>
>
> >**W2**.  The presentation can be improved:
> It needs to be clarified how training-free works. I think the authors should more explicitly describe it. My understanding is that training-free = 50 prototypes only (the second row in Table 4). Correct me if I am wrong.
> The description of the training process is also vague. Section 4 omits details on how prototype features are fine-tuned. It seems to me that the text encoder and the prompts are only used to initialize the prototypes. Correct me if I am wrong.
>
> **Re W2**. Thank you for your valuable feedback regarding the clarity of our training-free approach and the overall training process. We agree that these aspects require more explicit descriptions and have revised our paper accordingly.
>
> The training-free version indeed utilizes only 50 prototypes. These prototypes are initialized by the CLIP text encoder using specifically designed prompts that correspond to the categories or classes in our dataset. This initialization is critical as it provides a foundation for our zero-shot classification approach. This training-free method contrasts with our trained approach, where after the initial prototype initialization, we further fine-tune the prototype features. The fine-tuning process involves adjusting these features to better align with the specific nuances of our dataset, thereby improving the classification performance.
>
> To address the reviewer’s concern about the vagueness in our training description, we have expanded Section 4 to include a detailed account of this process. This includes how the prototypes are initialized, the role of the text encoder and prompts in this initialization, and the subsequent fine-tuning steps, if applicable. Specifically, we explain how the text encoder and prompts serve as a starting point, and how image samples from our dataset are used to refine the prototypes for improved accuracy and applicability.

---

> > ### Author Response · Authors · 2023-11-17
> > **Initial Response to Reviewer tgxP  [Cont']**
> >
> > >**Q1**. The authors primarily experimented with one CLIP model. It is unclear if this method can work with different CLIP variants, open-sourced CLIP replication or other VLM models. I'm curious if changing the model architecture, training data, or VLM format would yield different results.
> > While the method appears to be general, I'm concerned about it potentially "overfitting" to a specific model and dataset.
> >
> > **Re Q1**. Thank you for highlighting the importance of testing our method's generalizability across different CLIP variants and other VLMs. We recognize the concern regarding potential overfitting to a specific model or dataset. To address this, we conducted extensive experiments with various CLIP models and other VLMs, as detailed in the revised table below.
> >
> > | Variants | Set | CLIP  |CLIP-80| APPLe* | APPLe |
> > |:---------|:----|:-----:|:-----:|:-------:|:-----:|
> > | ViT-B/16 | Base| 72.43 | 73.56 | 74.62  | 78.17 |
> > |          | New | 68.14 | 69.97 | 71.94  | 72.12 |
> > |          | HM  | 70.22 | 71.72 | 73.26  | 75.02 |
> > | ViT-B/32 | Base| 67.43 | 67.52 | 69.40  | 72.66 |
> > |          | New | 64.04 | 65.84 | 67.78  | 67.83 |
> > |          | HM  | 65.69 | 66.67 | 68.58  | 70.16 |
> > | ViT-L/14 | Base| 79.20 | 79.96 | 81.03  | 83.51 |
> > |          | New | 74.02 | 76.43 | 78.06  | 78.13 |
> > |          | HM  | 76.52 | 78.16 | 79.52  | 80.73 |
> > |ViT-L/14@336|Base|80.25 | 81.04 | 82.00  | 84.26 |
> > |          | New | 75.50 | 77.60 | 79.08  | 79.15 |
> > |          | HM  | 76.52 | 78.16 | 80.51  | 81.63 |
> > |LAION ViT-B-32|Base|70.01| 70.07| 70.53  | 73.45 |
> > |          | New | 69.06 | 69.66 | 70.13  | 70.22 |
> > |          | HM  | 69.53 | 69.86 | 70.33  | 71.80 |
> > |BLIP ViT-B| Base| 43.63 | 50.10 | 54.40  | 67.56 |
> > |          | New | 48.42 | 59.16 | 63.38  | 66.28 |
> > |          | HM  | 45.90 | 54.25 | 58.55  | 66.91 |
> >
> > Our method's consistent performance improvement across different architectures, including ViT-B/16, ViT-B/32, ViT-L/16, ViT-L/16@336, LAION's CLIP replication [1], and the BLIP model [2], attests to its robustness. These results underscore our method's adaptability to different model architectures and training datasets.
> >
> > [1] Schuhmann, Christoph, et al. "Laion-5b: An open large-scale dataset for training next generation image-text models." Advances in Neural Information Processing Systems 35 (2022): 25278-25294.
> > [2] Li, Junnan, et al. "Blip: Bootstrapping language-image pre-training for unified vision-language understanding and generation." International Conference on Machine Learning. PMLR, 2022.
> >
> > >**Q2**. How does the training process for cross-dataset transfer work? When training on the source data (e.g., ImageNet), the model learns prototype features for ImageNet classes and adaptive attention weights for them. How does this transfer to target datasets where prototypes and attention weights remain untouched during fine-tuning?
> >
> > **Re Q2**. In response to your question about the training process for cross-dataset transfer, let me provide a more detailed explanation. When fine-tuning on base classes from the source dataset ImageNet, we include prototypes for both base and new classes. This inclusion allows the model to learn and optimize features that are not only specific to the base classes but also relevant to the new classes in the target dataset.
> >
> > The fine-tuning process involves optimizing the similarities between training samples and target datasets, particularly through the use of loss functions $L_{max}$ and $L_{dec}$. $L_{max}$ focuses on maximizing the relevant features for each class, while $L_{dec}$ works on decreasing the influence of irrelevant features. This dual approach ensures that the model not only retains the essential characteristics learned from the base classes but also adapts effectively to the nuances of the new classes in the target dataset.
> >
> > For example, when transferring from ImageNet to a more specialized dataset like OxfordPets, our model uses the learned prototypes and attention weights as a starting point. It then fine-tunes these components, guided by $L_{max}$ and $L_{dec}$, to better represent and distinguish the unique features of pet classes.

---

> > > ### Author Response · Authors · 2023-11-17
> > > **Initial Response to Reviewer tgxP [Cont']**
> > >
> > > >**Q3**. Could you clarify the importance of the quality of prompts used in the experiments? What would happen if we used GPT-4 to generate the prompts? How does the quality of the input prompt to the GPT model impact the final performance?
> > >
> > >
> > > **Re Q3**. Thank you for your concern about the impact of prompt quality on our model's performance. The quality of prompts is a critical factor in our experiments, as they directly influence the model's ability to accurately interpret and classify images based on textual descriptions.
> > >
> > > In our experiments, we observed that the performance with prompts generated by GPT-3 is relatively consistent with those generated by GPT-4. This could be attributed to the fact that generating category descriptions for our tasks may not require advanced reasoning capabilities, a domain where GPT-4 has more significant improvements over GPT-3. Hence, GPT-3's capacity appears to be sufficient for this specific task.
> > >
> > > Regarding the column labeled 'mixed1' in our table, it represents a mix of prompts generated by both GPT-3 and GPT-4. We included this to examine the impact of using a heterogeneous set of prompts on model performance. The results indicate that there is not a significant deviation in performance when using mixed-quality prompts compared to those generated solely by GPT-3 or GPT-4.
> > >
> > > We acknowledge the limitations in our current testing due to the time constraints of the rebuttal period. In the next version of our paper, we plan to conduct more comprehensive testing across all 11 datasets to demonstrate the consistency of performance regardless of the prompt generation source. This will provide a more complete picture of how different qualities of prompts impact the overall effectiveness of our model.
> > >
> > >
> > > | Datasets | Set | GPT-3 | GPT-4 | mixed1|
> > > |:---------|:----|:-----:|:-----:|:-----:|
> > > | DTD      | Base| 82.41 | 83.10 | 81.95 |
> > > |          | New | 69.57 | 70.41 | 70.72 |
> > > |          | HM  | 75.45 | 76.23 | 75.92 |
> > > |OxfordPets| Base| 95.64 | 95.43 | 95.69 |
> > > |          | New | 98.04 | 97.93 | 98.21 |
> > > |          | HM  | 96.83 | 96.66 | 96.93 |
> > > | Aircraft | Base| 44.66 | 45.14 | 45.20 |
> > > |          | New | 43.13 | 39.41 | 41.81 |
> > > |          | HM  | 43.88 | 42.08 | 43.44 |
> > >
> > >
> > > >**Q4**. Although the authors claim that fine-tuning the prompt textual features does not lead to overfitting issues, there are no ablations on the performance of training with frozen prototype features to demonstrate whether fine-tuning the prototype is necessary.
> > >
> > > **R4**. Thank you for raising the important question regarding the fine-tuning of prompt textual features and its relation to overfitting. We conducted an ablation study to specifically address this concern, focusing on the performance impact of training with frozen prototype features.
> > >
> > > The results, as shown in the revised table below, reveal that when the prompt textual features are not fine-tuned (i.e., prototype features are frozen), the performance improvement is less pronounced compared to when these features are fine-tuned. Specifically, the 'APPLe frozen features' method shows a marginal increase in performance for the 'New' class but a noticeable decrease in the 'Base' and 'Harmonic Mean (HM)' metrics, compared to the fully fine-tuned 'APPLe' method. This observation underscores the significance of fine-tuning the prompt textual features in our approach. Fine-tuning allows the model to adapt the prototype features more closely to the specific characteristics of the dataset, enhancing its ability to accurately classify images. Moreover, our results suggest that this fine-tuning does not lead to overfitting. This is evident from the consistent improvement across different classes without a trade-off between 'Base' and 'New' class performance.
> > >
> > > | Methods| APPLe* | APPLe |APPLe frozen features|
> > > |:-------|:-----:|:------:|:------:|
> > > | Base   | 74.62 | 78.17 | 75.46  |
> > > | New    | 71.94 | 72.12 | 72.07  |
> > > | HM     | 73.26 | 75.02 | 73.73  |

---

> > > > ### Author Response · Authors · 2023-11-17
> > > > **Initial Response to Reviewer tgxP [Cont']**
> > > >
> > > > >**Q5**. In Equation 7, the authors selected a method that can balance between all prototypes and the closest prototypes. Are there other balancing methods, such as using the Boltzmann operator or logsumexp, that could be considered?
> > > >
> > > > **Re Q5**. Thank you for suggesting the exploration of alternative balancing methods like the Boltzmann operator and logsumexp in Equation 7. We conducted additional experiments on ImageNet using these methods and present the results in the revised table below. We tested different Boltzmann temperatures (T=20, 10, 5) to understand their impact on performance. Our findings show that the mean/max balancing method we originally used outperforms both the Boltzmann operator and logsumexp. We hypothesize that this superiority is attributed to the specific way mean/max balancing integrates information from all prototypes while emphasizing the most relevant ones, which seems particularly effective for our model and dataset.
> > > >
> > > > Regarding the Boltzmann operator, we observed that varying the temperature has a noticeable impact on performance. Lower temperatures (T=5) led to results closer to our method. However, none of the temperatures tested could match the performance achieved by our mean/max balancing. The performance with logsumexp, especially in the 'Base' class, was notably lower. We will include these findings in our revised paper, along with a comprehensive discussion of these observations.
> > > >
> > > > | Methods| Mean/Max Balancing| Boltzmann T=20 | T = 10 |  T = 5 | Logsumexp |
> > > > |:-------|:-----:|:------:|:------:|:------:|:------:|
> > > > | Base   | 78.17 | 76.96 | 77.35  | 77.30 | 70.48 |
> > > > | New    | 72.12 | 71.96 | 71.96  | 72.07 | 71.64 |
> > > > | HM     | 75.02 | 74.27 | 74.56  | 74.59 | 73.73 |
> > > >
> > > > >**Q6**. While the authors aim for diverse prompts, it might be interesting to fix the prototype and only train the attention weights and forcing the attention weights to be a multinomial distribution with low entropy. This would be essentially learning to select the best prototype. It would be interesting to see if GPT-3 can produce a better single prompt than the hand-designed prompts used in CLIP.
> > > >
> > > > **Re Q6**. Thank you for the intriguing suggestion to experiment with training attention weights only and using low entropy to essentially select the best prototype. We conducted this experiment on ImageNet and compared the performance with CLIP’s hand-designed prompts and our APPLe* model.
> > > >
> > > > As shown in the table below, the performance using the best prototype selected by the model was inferior to that of the hand-designed prompts used in CLIP. This outcome suggests that relying on a single prototype, even if it's the 'best' as determined by the model, may not effectively capture the diverse and complex nature of objects in images. Our results indicate that a single prototype is often biased towards a specific representation of an object, which limits its generalizability.
> > > >
> > > > | Methods| CLIP | CLIP-closest prompt | APPLe* |
> > > > |:-------|:-----:|:------:|:------:|
> > > > | Base   | 72.43 | 71.84 | 74.62  |
> > > > | New    | 68.14 | 67.90 | 71.94  |
> > > > | HM     | 70.22 | 69.81 | 73.26  |
> > > >
> > > > For example, as highlighted in Figure 1 of our paper, each prompt for an apple pie depicts a particular state or aspect of the pie, such as a round pie or a slice on a plate. While each prompt is accurate in its description, none can encompass all the variations of apple pies alone. This specificity is where the limitation lies. It becomes challenging for a single, even well-crafted prototype to represent the breadth of variations that an object can have in the real world.
> > > >
> > > > >**Q7**.  Have the authors attempted to use the embedding ensembling method used in CLIP?
> > > >
> > > > **Re Q7**. Thank you for providing this valuable suggestion. We have provided embedding ensembling method results in Re W1.
> > > >
> > > >
> > > > >**Q8**. In Equation 7, stating the "average cosine similarity" is not entirely accurate because the cosine similarities are weighted by the attention weights.
> > > >
> > > > **Re Q8**. Thank you for pointing out the need for more precise terminology in describing our model's mechanics. We agree that the term 'average cosine similarity' in Equation 7 could potentially be misleading, as it does not fully capture the role of attention weights in the calculation. Therefore, we have revised our paper to use 'weighted average cosine similarity' in the relevant sections.
> > > >
> > > > >**Q9**. While the trend in Figure 5 is clear, it could be valuable to include settings with 0/1 and 1/0 to further illustrate the findings.
> > > >
> > > > **Re Q9**. We appreciate your suggestion to include settings with 0/1 and 1/0 ratios in Figure 5 to provide a more comprehensive understanding of our findings. We have revised Figure 5 accordingly, and the additional results are as follows:
> > > >
> > > > | Ratio | Base  |  New  |   HM  |
> > > > |:------|:------|:-----:|:-----:|
> > > > | 0/1   | 70.17 | 67.74 | 68.93 |
> > > > | 1/0   | 76.65 | 71.86 | 74.18 |

---

> > > > > ### Comment · Reviewer_tgxP · 2023-11-23
> > > > > **Re**
> > > > >
> > > > > Thank you for your responses.
> > > > >
> > > > > I have a follow-up on Q6.
> > > > > The result is interesting and it also shows that each of the authors' prompt are potentially more detailed and the prompts are more orthogonal to each other.
> > > > > I am wondering if there is still time to see how the "diversity" of the prompts affect the results, and could that be the reason why GPT-3 is better than GPT-4: instruction-tuned model may be less diverse compared to untuned foundation models.

---

> > > > > > ### Author Response · Authors · 2023-11-23
> > > > > > **Response to Follow-up Question on Q6**
> > > > > >
> > > > > > Dear Reviewer tgxP,
> > > > > >
> > > > > > We sincerely appreciate your follow-up question concerning Q6 and are grateful for your insightful perspective on the diversity of prompts. Your observation regarding the potential for GPT-4, as an instruction-tuned model, to generate less diverse prompts compared to the untuned foundation model GPT-3 is quite intriguing and appears to be accurate.
> > > > > >
> > > > > > In light of this, we are actively working to assess and provide detailed analysis on how this "diversity" factor impacts performance. We aim to incorporate this analysis and will update our findings accordingly ASAP.
> > > > > >
> > > > > > Thank you once again for your valuable input, which has significantly contributed to the depth and scope of our study.
> > > > > >
> > > > > > Best regards,
> > > > > >
> > > > > > The Authors

---

> > > > > > ### Author Response · Authors · 2023-11-23
> > > > > > **Impact of Prompt Diversity**
> > > > > >
> > > > > > Dear Reviewer tgxP,
> > > > > >
> > > > > > Thank you for your insightful suggestion regarding the evaluation of prompt diversity in our experiments. Based on your feedback, we have conducted an additional analysis to explore the impact of prompt diversity on model performance.
> > > > > >
> > > > > > From the existing prompts (50 prompts per class), we sampled 100 sets of prompts for ImageNet classes, with each set containing 10 prompts per class. To assess the diversity within these sets, we calculated the cosine similarity matrix for the textual features of each set and measured the standard deviation (std) of these matrices. We believe this standard deviation serves as a simple yet effective metric to gauge prompt diversity.
> > > > > >
> > > > > > From these 100 sets, we selected 10 sets with the most distinct standard deviations, representing varying levels of prompt diversity. We then evaluated the performance of our model using these 10 sets. The results, as shown in the table below, indicate a clear trend: higher diversity (higher std) correlates with improved performance. This finding suggests that increased diversity in prompts can enhance the model's effectiveness. As the sampling pool is quite limited, each class only have 50 prompts, we believe if we generate a larger pool, the trend will be more obvious.
> > > > > >
> > > > > > We plan to incorporate these findings into our experiments, as they provide valuable insights into prompt selection strategies, particularly when a larger pool of prompts is available. We believe this addition will further strengthen the contributions of our work.
> > > > > >
> > > > > > With the discussion phase deadline approaching, we remain open and eager to address any further questions or suggestions you may have.
> > > > > >
> > > > > > Best regards,
> > > > > >
> > > > > > The Authors
> > > > > >
> > > > > > | Diversity Rank (std) |     10(0.1260)    |     9(0.1272)    |     8(0.1278)   |      7(0.1281)    |    6(0.1282)    |      5(0.1284)  |     4(0.1287)    |      3(0.1288)   |     2(0.1291)   |     1(0.1300)   |
> > > > > > |:--------------------|:--------:|:-------:|:--------|:--------:|:-------:|:--------|:--------:|:-------:|:--------|:--------:|
> > > > > > |        Base         | 72.43 |  72.93|   72.70|  72.71|  72.20|  72.41 | 72.54| 72.92|  72.24 | 73.05
> > > > > > |        New          | 69.80 |  69.75|   69.80|  70.04|  69.63|  70.32 | 69.96| 69.97|  70.04 | 70.31
> > > > > > |         HM           | 71.09 |  71.30|   71.22|  71.35|  70.89|  71.35 | 71.23| 71.41| 71.12  | 71.65

---

> ### Author Response · Authors · 2023-11-23
> **Follow-up questions before the discussion phase ends**
>
> Dear Reviewer tgxP,
>
> First and foremost, we would like to extend our gratitude for the positive feedback you have provided on our work. Your kind words are both encouraging and appreciated.
>
> In response to the points of weakness you highlighted, we have previously submitted detailed, point-by-point responses. We hope that these have satisfactorily addressed the concerns about our paper.
>
> As the discussion phase is approaching its conclusion in the next 12 hours, we eagerly anticipate any additional feedback you might have on our responses. Should there be any further questions or clarifications needed, we are ready and willing to provide additional information in a timely manner.
>
> Thank you once again for your valuable input and the time you have invested in reviewing our work.
>
> Best regards,
> The Authors

---

### Author Response · Authors · 2023-11-17
**A Summary of Revisions**

We would like to thank all the reviewers for their time and effort providing constructive feedback. To make our method evaluation stronger and improve the readability and clarity of the paper, we have revised both the main text and supplementary material based on the comments, which are highlighted in blue. Building on reviewers’ comments as well as expanding on the discussion points in our responses, we have made the following changes to the manuscript:

**Experiments**: we have included more experimental results in **few-shot learning** in Section A.4, **comparison with the generic 80 prompts** in Section A.5, **generalization across different CLIP variants and other VLMs** in Section A.6, **quality of prompt generation impact** in Section A.7, **more prototype calibration strategies** in Section A.8, **best prototype selection performance** in Section A.9, **0/1 and 1/0 prototype calibration ratio** in Figure 5, **ablation study with frozen textual features** in Table 4.

**Related Work**: we have included the comparison to the related work [1,2] and discuss the differences of our work to them.

>[1] Menon, Sachit, and Carl Vondrick. "Visual Classification via Description from Large Language Models." ICLR 2023.
>[2] Pratt S, Covert I, Liu R, et al. What does a platypus look like? generating customized prompts for zero-shot image classification. ICCV 2023.

**Clarifications**: we have clarified the training-free and training-based versions in Section 4 Training and Inference. We have elaborated on the cross-dataset transfer setting in Section 5.

Please let us know if there are any questions or suggestions. Thanks again for the insightful suggestions provided. We do hope our responses and the updated manuscript can help ease the concerns.

---

### Author Response · Authors · 2023-11-22
**Happy to take follow-up questions before the author-reviewer discussion closes**

Dear reviewers and AC,

Thank all for your time and insightful suggestions. As the first phase of the discussion period is about to end today, we are wondering whether anyone has further questions or comments on our revised manuscript and responses. It is very important for us to hear the feedback from you and address the concerns by today :)

Thanks again in advances!

Authors

---

### Meta-Review · Area_Chair_GWJL · 2023-12-04

**Metareview:**

This paper received mixed reviews initially. The raised issues include insufficient experimental validations, limited performance gain by the proposed loss function, and unclear technical presentations. During the rebuttal phase, the authors addressed the raised issues by providing technical explanations and additional experiments. However, there are still concerns regarding the technical contributions (e.g., the weight and loss designs by fKhC) that are not solved completely. Overall, the reviewers are lukewarm about the current submission and not quite supportive of reporting. After discussion with AC panels, the current submission shall undergo a major revision to emphasize the proposed contributions. Also, recent works regarding prototype learning shall be referred to and acknowledged.

[a]. Automatically Discovering Novel Visual Categories With Adaptive Prototype Learning. Zhang et al. PAMI 2023.
[b]. Evolving Semantic Prototype Improves Generative Zero-Shot Learning. Chen et al. ICML 2023.
[c]. Adaptive Prototype Learning and Allocation for Few-Shot Segmentation. Li et al. CVPR 2021

**Justification For Why Not Higher Score:**

Incremental research upon prototype learning.

**Justification For Why Not Lower Score:**

N/A

---

### Decision · Program_Chairs · 2024-01-16

Reject